# A class of organic cages featuring twin cavities

Zhenyu Yang [1,4], Chunyang Yu [1,4], Junjie Ding[1], Lihua Chen[1], Huiyu Liu[2], Yangzhi Ye[2], Pan Li[1], Jiaolong Chen[1], Kim Jiayi Wu[3], Qiang-Yu Zhu[1], Yu-Quan Zhao[1], Xiaoning Liu[1], Xiaodong Zhuang [1] & Shaodong Zhang [1✉]

A variety of organic cages with different geometries have been developed during the last decade, most of them exhibiting a single cavity. In contrast, the number of organic cages featuring a pair of cavities remains scarce. These structures may pave the way towards novel porous materials with emergent properties and functions. We herein report on rational design of a three-dimensional hexaformyl precursor **1**, which exhibits two types of conformers, *i.e.* Conformer-1 and -2, with different cleft positions and sizes. Aided by molecular dynamics simulations, we select two triamino conformation capturers (denoted CC). Small-sized CC-1 selectively capture Conformer-1 by matching its cleft size, while the large-sized CC-2 is able to match and capture both conformers. This strategy allows the formation of three compounds with twin cavities, which we coin diphane. The self-assembly of diphane units results in superstructures with tunable proton conductivity, which reaches up to $1.37 \times 10^{-5}$ S cm$^{-1}$.

[1] School of Chemistry and Chemical Engineering, Frontiers Science Center for Transformative Molecules, Shanghai Jiao Tong University, 800 Dongchuan Road, 200240 Shanghai, China. [2] School of Physical Science and Technology, ShanghaiTech University, 393 Huaxia Middle Road, 200120 Shanghai, China. [3] School of Chemistry, the University of Edinburgh, W. Mains Road, Edinburgh EH9 3FJ, UK. [4]These authors contributed equally: Zhenyu Yang, Chunyang Yu. ✉email: sdzhang@sjtu.edu.cn

Covalent organic cages with well-defined intrinsic porosity have attracted increasing attention for the last decade[1–8]. Their internal cavity and external channels have been applied for selective recognition/separation[2,3,9], catalysis[10,11], sensing[12], and so on. A variety of cages with different three-dimensional shapes have been elaborated, most of which exhibit a single cavity, with the exception of dumbbell-shaped cages with twin cavities recently reported by separate works of Li and Stang[13] and Cooper[14]. Besides, cages with different geometries can be employed as powerful building blocks in search of novel supramolecular materials with emergent properties that are otherwise inaccessible by conventional molecules;[15] however, this has been largely overlooked.

Our group has been focusing on the elaboration[16,17] and hierarchical self-assembly[18,19] of covalent organic cages prepared by dynamic covalent chemistry (DCC)[7,8]. Due to the dynamic and reversible nature of DCC, it allows error correction to yield the thermodynamic products[20,21]. This feature prompts us to rationally design a three-dimensional (3D) model compound **1** with molecular dynamics simulations (left column, Fig. 1). This conformationally labile molecule **1** with six formyl groups exhibits all instantly interchanging conformations that fall into two distinct types: Conformer-1 with two clefts located below and above the central arene, in each cleft three formyl groups tending to point inward; Conformer-2 with a pair of left and right clefts, where each three formyl moieties are outward positioned. As the formyl groups are permanently moving, the region with the probability of containing formyl moieties forms a circular ring within each cleft, and the range of the ring is different for Conformer-1 and -2 (middle column, Fig. 1). By virtue of their low isomerization barrier, their identification poses a formidable challenge for conventional methods with low time-scale resolution, such as nuclear magnetic resonance (NMR) and infrared (IR) spectroscopies. X-ray crystallography is a powerful method to unambiguously determine the absolute conformation of a molecule, but it often only distinguishes the conformer that provides the most efficient molecular packing[22].

In this work, we develop a size-matching strategy by choosing two triamino conformation capturers (abbreviated CC), i.e., CC-1 and -2 with different sizes. Through defect-checking DCC, these two molecular capturers can chemically capture Conformer-1 and -2 (right column, Fig. 1), which allows their facile differentiation by conventional NMR and X-ray analysis, and also leads to the formation of three cage-like compounds with twin cavities (see below)[13,14], which we coin diphane. Moreover, these three diphanes self-assemble into similar superstructures. Depending on the shape and width of the intermolecular channels, they exhibit the dramatic different capacity of proton conduction, which is distinct by an order of magnitude of $10^4$.

## Results

### Design and characterization of the model compound.
As briefly mentioned above, the region with the probability of containing formyl groups varies within the cleft of each conformer are determined by molecular dynamics simulations (see detailed MD simulation in Supplementary Fig. 10), which shows that molecule **1** exists infinite forms of conformers via all possible C–C single bond rotations, of which C–C bonds a are much more liable to rotate than C–C bonds b, as the latter has a higher rotational energy barrier (Supplementary Fig. 9). It, therefore, leads to the formation of two types of conformers, Conformer-1 and -2 (Figs. 1 and 2a). It reveals that Conformer-1 exhibits two clefts located below and above the central arene, in each of which three formyl groups tend to point inward (left, Fig. 2a); while in Conformer-2 each three formyl moieties are outward positioned

(right, Fig. 2a). The energy difference of the two conformers is extremely low (ca. $0.1 \, \text{kJ} \, \text{mol}^{-1}$, Supplementary Table 1 and Supplementary Fig. 9), indicating their fast interconversion, which could have been otherwise challenging to be distinguished by conventional analytical tools such as variable-temperature NMR[23,24] and time-resolved spectroscopies[25,26].

Accordingly, molecule **1** is synthesized by $\text{Pd(PPh}_3)_4$ catalyzed Suzuki–Miyaura cross-coupling reaction using commercially available molecule (2-formyl phenyl)boronic acid as reactant (51% five-step overall yield, see Supplementary Fig. 1 for details). The rapid averaging of its possible conformers is confirmed by variable temperature $^1\text{H}$ NMR spectroscopy in $\text{CDCl}_3$, with symmetric splitting patterns of all protons even at 223 K (Fig. 2b). Its single crystals suitable for X-ray crystallography are obtained by slow evaporation of solvent from its chloroform solution, and molecule **1** only exists the conformation corresponding to Conformer-1 in the crystal structure (Fig. 2c), which is common for the conformationally labile compounds that often crystallize into the most efficient molecular packing[22].

### Conformation capturing with diphane formation.
As the framework of cages can experience complex motions in space—including continuous bending, twisting, stretching, and shrinking, they also provide an excellent platform for the study of molecular conformation and topology[21]. By taking advantage of the dynamic and reversible nature of DCC, and facilitated by DFT calculation (vide infra), we choose CC-1 and -2 with flexible amine-capped chains and different sizes to tentatively capture the above-mentioned conformations of molecule **1** by error-checking imine formation (Fig. 3).

We first use the small-size CC-1, i.e., tris(2-aminoethyl)amine, to react with molecule **1** in the presence of $\text{Sc(OTf)}_3$ with a molar ratio of CC-1:**1**:$\text{Sc(OTf)}_3 = 2:1:0.6$ in chloroform, and the reaction mixture is then subjected to reduction with $\text{NaHB(OAc)}_3$ (15 equiv.), yielding the amine-containing product for easy purification (64% two-step overall yield, see Supplementary Fig. 4 for details). The resulting product is examined by matrix-assisted laser desorption/ionization time-of-flight mass spectrometry (MALDI-TOF MS), $^1\text{H}$ and COSY NMR, and single-crystal X-ray diffraction (SC-XRD). MALDI-TOF MS shows a single ion peak at $m/z$ 1384.7883, corresponding to the expected product with the formula $\text{C}_{98}\text{H}_{95}\text{N}_8$ ($[\text{M} + \text{H}]^+$ calcd for 1384.7708).

The single crystals suitable for X-ray analysis are obtained by slow evaporation of endo-[1,2,4]diphane solution in THF with additional trifluoroacetic acid (TFA) for better solubility. endo-[1,2,4]Diphane crystallizes into triclinic space group $P\bar{1}$ and unambiguously reveals only one isomer corresponding to Conformer-1 (Fig. 4a), in which the central arene is shared by two identical cavities. As this twin-cavity cage-like compound is composed of aromatic units and aliphatic chains, we, therefore, name it diphane. This molecule is given a systematic name endo-[1,2,4]diphane, where "endo" indicates the central arene is embedded within the twin cavities, "1" refers to the number of central benzene rings, "2" denotes the number of aromatic units of every rigid arm, and "4" counts the backbone atoms forming each flexible chain. Due to the significant intramolecular strain, the aliphatic chains are twisted in an irregular fashion. The twisting of these three chains is highly uneven (Fig. 4b), in agreement with the complex yet distinct splitting of each proton in the $^1\text{H}$ NMR spectrum (Fig. 4c). In addition, the X-ray structure of endo-[1,2,4]diphane shows that the tertiary amines on the aliphatic chains are not protonated, even with the presence of TFA. This is presumably due to the steric hindrance of the irregularly stretched amine moiety caused by the rigid strain conformation, as well as the electrostatic repulsion of the

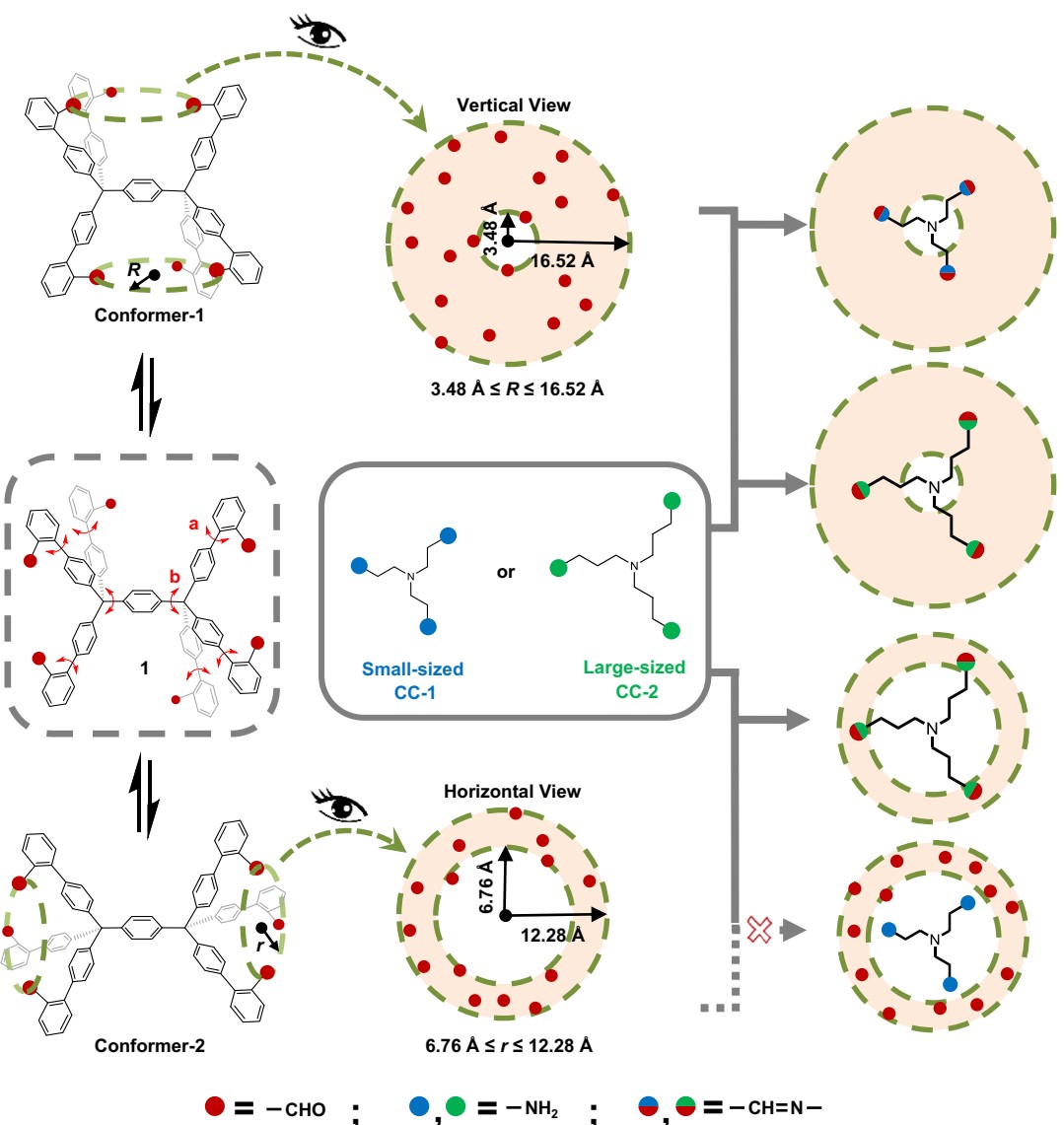

**Fig. 1 Schematic representation of size-matching strategy for facile identification of two types of conformations of model compound 1 with six formyl groups presented with red dots.** The three-dimensional molecule **1** experiences spontaneous interconversion between Conformer-1 with a pair of upper and lower clefts and Conformer-2 with a pair of left and right clefts (left column), of which C–C bonds a are much more liable to rotate than C–C bonds b, as the latter has a higher rotational energy barrier determined by relaxed potential energy scan (Supplementary Fig. 9). As revealed by molecular dynamics simulations, the permanently moving formyl groups are spotted within a projected circular ring with pink color (middle column), which is used to quantify the range covered by permanently moving formyl groups in each cleft. Considering the inner circle of the ring, i.e., the smallest possible size of the cleft, is different for Conformer-1 and -2, a small-sized CC-1 (with the amines presented with blue dots) and large-sized CC-2 (with the amines presented with green dots) are selected as size-matching conformational capturers (denoted as CC) by molecular dynamics simulations. Small-sized CC-1 exclusively captures and identifies Conformer-1, and large-sized CC-2 captures both Conformer-1 and -2 (right column).

neighboring TFA molecules (Supplementary Fig. 12). These results, therefore, indicate that our tethering strategy with conformer capturers provides an efficient way to "visualize" the flexible conformers in solution, with an added bonus of discovering the 3D structures with appealing topology with tunable materials properties (vide infra).

On the other hand, we fail to capture Conformer-2 by the formation of hypothetical *exo*-[1,2,4]diphane (see above), where "*exo*" indicates the central arene located outside the two compositional cages. This result therefore confirms the efficacy of our size-matching strategy, which theoretically rules out the formation of *exo*-[1,2,4]diphane, as the maximally stretched amines of CC-1 still cannot reach the three formyl groups within each cleft of Conformer-2 (vide supra, Fig. 1 and Supplementary Fig. 10).

We then employ the large-sized conformational capturer CC-2, namely tris(3-aminopropyl)amine, with one more carbon unit on each chain as compared to CC-1. The stoichiometric imination between CC-2 and molecule **1** followed by subsequent reduction yields two isolated products with the same ion peak at *m/z* of 1468.8496, corresponding to the expected diphane isomers with the formula $C_{104}H_{107}N_8$ ($[M + H]^+$ calcd for 1468.8647). To our delight, these two products are identified as *endo*-[1,2,5]diphane (59% two-step overall yield) and *exo*-[1,2,5]diphane (16% two-step overall yield) by SC-XRD and NMR spectroscopy (Fig. 4d–f, g–i, respectively, see Supplementary Fig. 5 for details). *endo*-[1,2,5]Diphane adopts similar conformation with *endo*-[1,2,4] diphane (Fig. 4d–f). On the other hand, *exo*-[1,2,5]diphane exhibits a dumbbell-shaped structure[13,14], with the middle arene

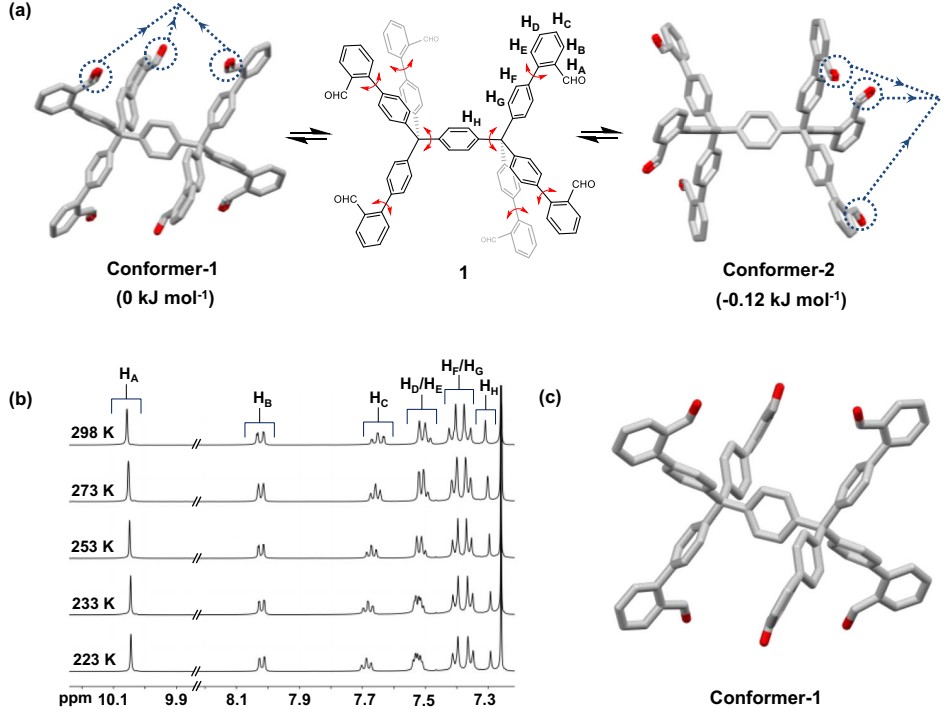

**Fig. 2 Conformational interconversion of molecule 1 and its identification by NMR spectroscopy and SC-XRD. a** Fast interconversion of Conformer-1 and -2 with extremely low isomerization barrier. Free energy of each isomer is determined by DFT calculation (B3LYP-D3 in PCM with chloroform as solvent). **b** Partial variable-temperature $^1$H NMR spectra (500 MHz) of molecule **1** in CDCl$_3$, showing rapid averaging of its possible conformers even at low temperature. The broadening of peaks of molecule **1** at low temperatures is due to its decreased solubility. **c** Molecular structure of molecule **1** revealed by SC-XRD, only able to identify Conformer-1 in solid-state. Solvent molecules and hydrogen atoms are omitted for clarity. Carbon atoms are labeled in gray, and oxygen in red. Dashed lines are used to facilitate the inspection of the orientation of formyl groups.

connecting two individual cavities. In contrary to its *endo*-isomer, *exo*-[1,2,5]diphane manifests a highly symmetric conformation, as each aromatic arm and flexible chain is twisted and stretched in the same manner (Fig. 4h). This is also confirmed by the simple splitting of each proton in $^1$H NMR spectrum (Fig. 4i).

**Mechanistic study of conformation capturing and diphane formation.** In order to investigate the mechanism of exclusive capturing of Conformer-1 with small-sized CC-1 in solution, we designed the control experiments (Fig. 5). The triformyl model compound 2 is the half truncated version of molecule **1**, which eliminates the formation possibility related to the conformation of *endo*-[1,2,4]diphane or *endo*-[1,2,5]diphane. It, therefore, enables the direct inspection of the size effect of conformation capturers CC-1 and CC-2. When small-sized CC-1 is used, the formation of Cage-1 can only be detected by mass spectroscopy but not by NMR, which is presented here with its optimized structure by DFT calculation (Fig. 5); while Cage-2 is formed with large-sized CC-2 under the same reaction conditions (see Supplementary Fig. 3 for details). The single crystals of Cage-2 confirms its expected structure. The strain energies (SE) of both products are also examined, and it shows that Cage-1 exhibits a much higher strain as compared to Cage-2 (48.00 vs. 36.96 kJ mol$^{-1}$, Fig. 5). Besides, the rigid arm of the optimized Cage-1 is significantly bent with a torsion angle of $\tau_1 = 174.66°$, while the arm of Cage-2 is much closer to the loose conformation with $\tau_1 = 176.57°$. This high strain of Cage-1 presumably prevents its formation, which is also true as for *exo*-[1,2,4]diphane.

To gain further insight into the capturing process between Conformer-1 and -2 with the larger capturer CC-2, the kinetic experiment is carried out with $^1$H NMR at 298 K in CDCl$_3$ without reduction of the resulting imine bonds (Fig. 6a). Initially,

all possible conformers of molecule **1** interconvert rapidly in solution, and the $^1$H NMR spectrum displays average splitting before the addition of CC-2 at 0 h. Upon addition of CC-2, the intensities of H$_A$ and H$_B$ of molecule **1** decrease steadily with time until their complete disappearance after four hours. Meanwhile, two groups of characteristic peaks, respectively, colored in light blue and red, emerge and gradually increase until the plateau at 4 h.

The peaks in light blue represent the formation of the imine form of *endo*-[1,2,5]diphane, denoted *endo*-[1,2,5]diphane', of which H$_C$ is the proton of shiff base and H$_D$ the aromatic proton adjacent to H$_C$. The peaks in light red are assigned to the imine form of *exo*-[1,2,5]diphane, namely *exo*-[1,2,5]diphane', of which H$_{C'}$ corresponds to the proton of shiff base and H$_{D'}$ the aromatic proton in the immediate vicinity of H$_{C'}$. The evolution profile illustrates that the consumption of molecule **1** occurs simultaneously with the formation of two diphane isomers until the equilibrium, which has a product distribution of 75% *endo*- and 25% *exo*-[1,2,5]diphane', respectively. In addition, a parallel study shows that *endo*-[1,2,4]diphane' can be transformed to *endo*-[1,2,5]diphane' and *exo*-[1,2,5]diphane', when additional large-sized **CC-2** is charged into the reaction mixture (Supplementary Fig. 8). It clearly indicates that *endo*- and *exo*-[1,2,5]diphane' are thermodynamically more stable than *endo*-[1,2,4]diphane'.

These results are in line with DFT calculations of SE and torsion angle of each amine form of diphane (Supplementary Fig. 11). The SE of *exo*-[1,2,4]diphane is the highest with 102.13 kJ mol$^{-1}$, larger than 96.38 kJ mol$^{-1}$ for *endo*-[1,2,4]diphane, and *exo*-[1,2,4]diphane is significantly bent with the smallest torsion angle of $\tau_4 = 174.12°$, smaller than 175.04° for *endo*-[1,2,4]diphane. When using the large-sized CC-2, the SE values of the resulting diphanes are considerably lower, with

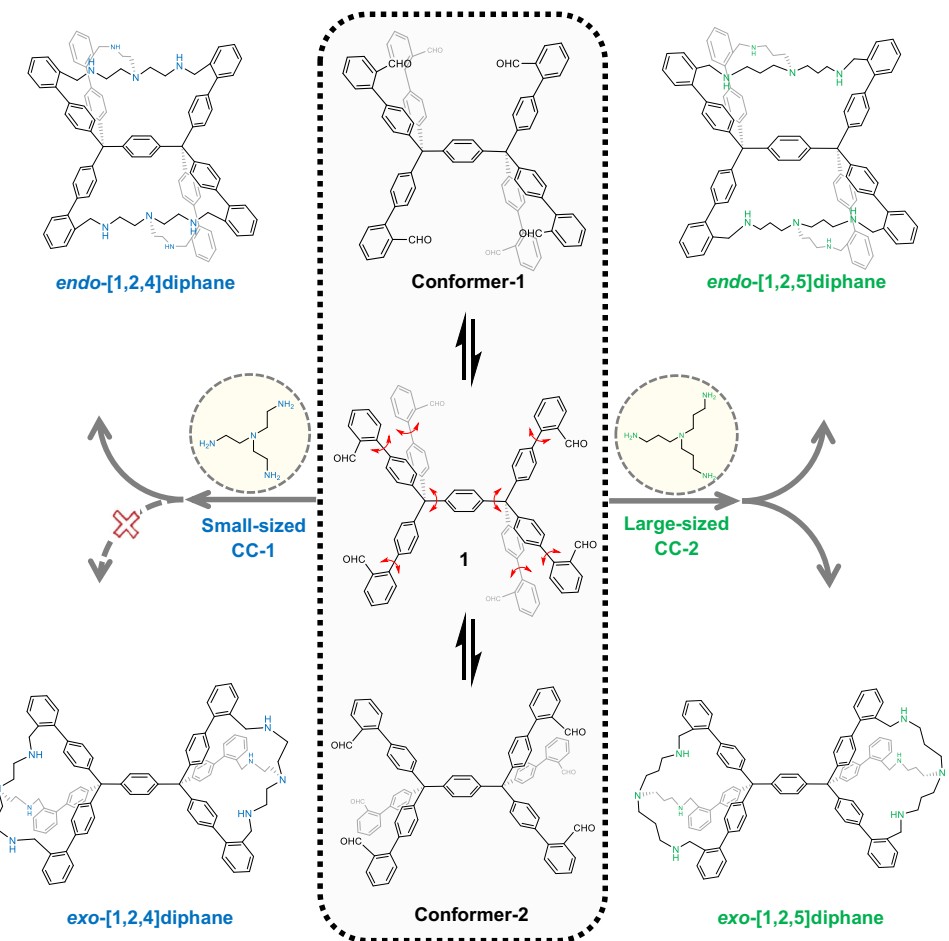

**Fig. 3 Conformation capturing of molecule 1 leading to the formation of three diphanes with twin cavities.** Schematic representation of conformation capturing of two conformers, i.e., Conformer-1 and -2 of model compound **1** among its infinite interconvertible conformers in solution, achieved by dynamic and reversible cycloimination and subsequent reduction. Small-sized conformer capturer CC-1 (presented in blue) exclusively captures Conformer-1, leading to the formation of *endo*-[1,2,4]diphane. Large-sized CC-2 (showed in green) captures both Conformer-1 and -2, yielding *endo*-[1,2,5]diphane and *exo*-[1,2,5]diphane, respectively.

67.33 kJ mol$^{-1}$ for *endo*-[1,2,5]diphane, and 77.00 kJ mol$^{-1}$ for *exo*-[1,2,5]diphane, respectively, echoed with their corresponding torsion angles.

Proton conduction is an essential procedure in biology and proton-exchange membrane fuel cells[27,28]. As our diphanes contain amine moieties, which can efficiently host protic acids and eventually crystallize into proton-conducting porous crystals. We, therefore, encapsulate TFAs into the crystalline superstructures of the three diphanes (denoted as diphanes–TFA) and evaluate their performances of proton conduction (Fig. 7). As determined by SC-XRD, *endo*-[1,2,4]diphane and *endo*-[1,2,5] diphane exhibit similar geometry, i.e. sandglass-shaped structure (Fig. 7a, c). However, their self-assembled superstructures are remarkably different. *endo*-[1,2,4]Diphane molecules are closely packed, and it provides no clear transportation channels for protons hopping (Fig. 7b), where TFA molecules are trapped within the intrinsic and extrinsic cavities of the diphanes. In contrast, *endo*-[1,2,5]diphanes self-assemble into a crystalline phase with ordered channels (highlighted with red column) that allow the transportation of protons (Fig. 7d). *exo*-[1,2,5]Diphane adopts a dumbbell-shaped geometry, and its corresponding superstructure also exhibits protons channels (Fig. 7f). Compared to the superstructure of *endo*-[1,2,5]diphane–TFA with ordered channel ($d_1 = 6.7$ Å), the diphane molecules in *exo*-[1,2,5] diphane–TFA are packed more tightly, and form zig-zag and

narrower channels ($d_2 = 4.7$ Å), which might hinder the transportation of protons.

We subsequently measure the proton conductivity of these superstructures at 303 and 333 K, respectively, with a relative humidity (RH) of 48% (Table 1). Regardless of the geometry of diphanes, their proton conductivity increases smoothly with temperature (Supplementary Table 7), in line with the behaviors of previously reported porous materials[29,30]. Although the exact location of water molecules within these superstructures is currently uncertain, their activation energies are lower than 0.4 eV, indicating the proton conduction was realized via the Grotthuss mechanism, where protons are hopping within the channels through interconnected hydrogen bonding[30,31]. In addition, the dynamics of diphanes–TFA are further investigated through various solid-state NMR (ssNMR) spectra[32,33]. It is found from the comparison of the single-pulse $^1$H MAS and dipolar-based 1D $^1$H DQ/SQ NMR spectra (Supplementary Fig. 22), where the protonic peaks of diphanes–TFA are almost suppressed, while the rigid protons remained in 1D $^1$H DQ/SQ NMR spectra. These results clearly indicate that the –NH$_2^+$ related protons of diphanes–TFA are highly mobile, whose dipolar interactions are remarkably reduced by molecular motions. Furthermore, the conductivities of diphanes are significantly enhanced by doping with additional TFA (Supplementary Fig. 17), where the increased number of protons acting

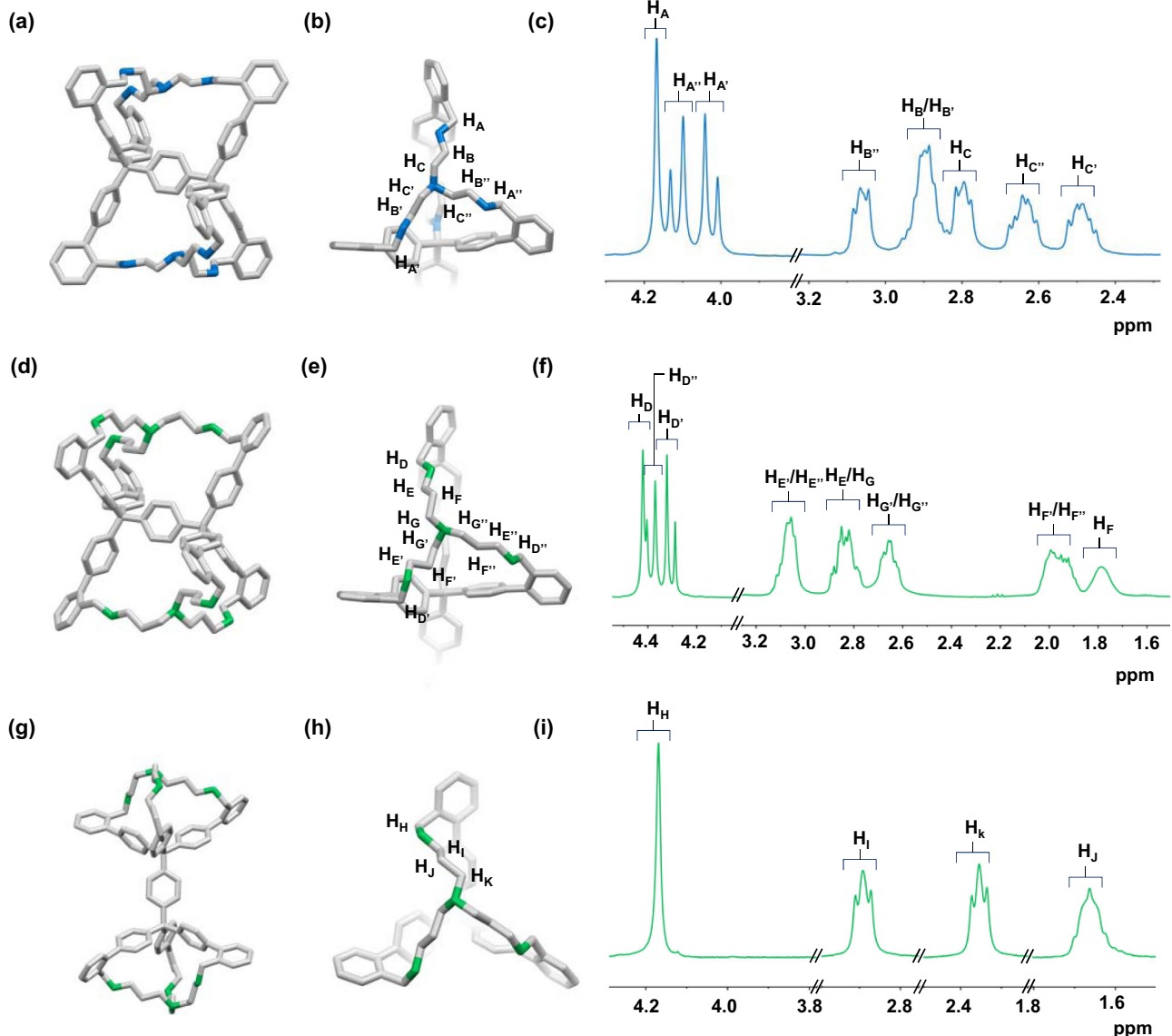

**Fig. 4 Molecular structures of three diphanes determined by SC-XRD and ¹H NMR spectroscopy.** Side views (**a**, **d**, **g**), truncated top views (**b**, **e**, **h**), and their corresponding partial ¹H NMR spectra (**c**, **f**, **i**) of *endo*-[1,2,4]diphane (in blue and gray), *endo*-[1,2,5]diphane (in green and gray) and *exo*-[1,2,5] diphane (in green and gray), respectively. Hydrogen atoms and solvent molecules are omitted for clarity. Carbon atoms are labeled in gray, and nitrogen in blue and green to distinguish the shorter and longer flexible chains, respectively.

as charge carriers are essential to the formation of excess protons in the hydrogen bond network, facilitating the proton conduction[34].

More importantly, the proton conductivity clearly depends on the packing of diphanes in their crystalline phases. The superstructure formed by *endo*-[1,2,4]diphanes–TFA without apparent channel exhibits the lowest proton conductivity at $2.44 \times 10^{-9}$ S cm⁻¹. This conductivity is significantly enhanced by the formation of proton channels, particularly for *endo*-[1,2,5] diphanes–TFA with the widest and most ordered channels, which reach up to $1.37 \times 10^{-5}$ S cm⁻¹, ~$1 \times 10^3$ times higher than bulk water[29,35].

## Discussion

In the current study, we develop a size-matching strategy to identify and distinguish the interconverting conformational isomers that are otherwise difficult to be distinguished by conventional analytical tools. The proof-of-concept hexaformyl model

compound **1** exhibits infinite conformers that fall into two types of conformers, namely Conformer-1 with a pair of upper and lower clefts and Conformer-2 with a pair of left and right clefts. As revealed by molecular dynamics simulations, the sizes of clefts are different for the two conformers. By making the most of the dynamic and reversible nature of cycloimination, we choose two triamino conformation capturers, i.e., CC-1 and -2 with different sizes to match and chemically tether the two conformers with different-sized clefts. It facilitates their retrospective differentiation by conventional NMR and X-ray analysis. This method allows us to discover three cage-like compounds with twin cavities, denoted diphanes. The sandglass-shaped *endo*-[1,2,4] diphane and *endo*-[1,2,5]diphane have two interconnected pockets by sharing a common aromatic vertex, while the dumbbell-like *exo*-[1,2,5]diphane exhibits two individual apertures connected by a joint benzene ring.

This method opens up avenues for the discovery of 3D molecules with exotic geometries and cavities. These molecules with unique shapes and configurations may be used as

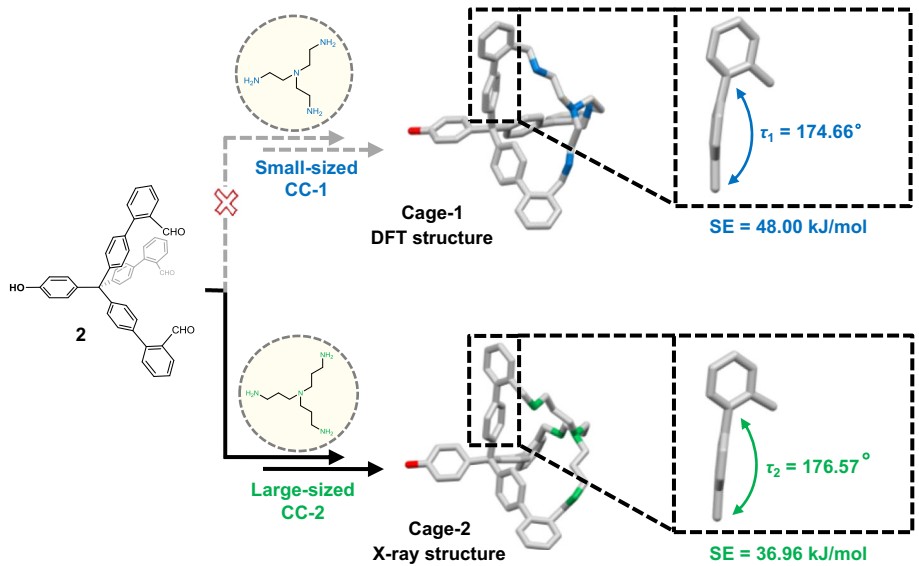

**Fig. 5 Inspection of size effect of conformation capturers CC-1 and CC-2 with model compound 2.** Small-sized CC-1 (blue) fails to form Cage-1, which is presented here with a geometry optimized structure (DFT, B3LYP-D3 in PCM with chloroform as solvent); While CC-2 (green) succeeds in yielding Cage-2, with its molecular structure determined by X-ray crystallography. Hydrogen atoms and solvent molecules are omitted for clarity. The insets show the enlarged rigid arm of the two cages, revealing their torsion angles and the corresponding strain energies, respectively.

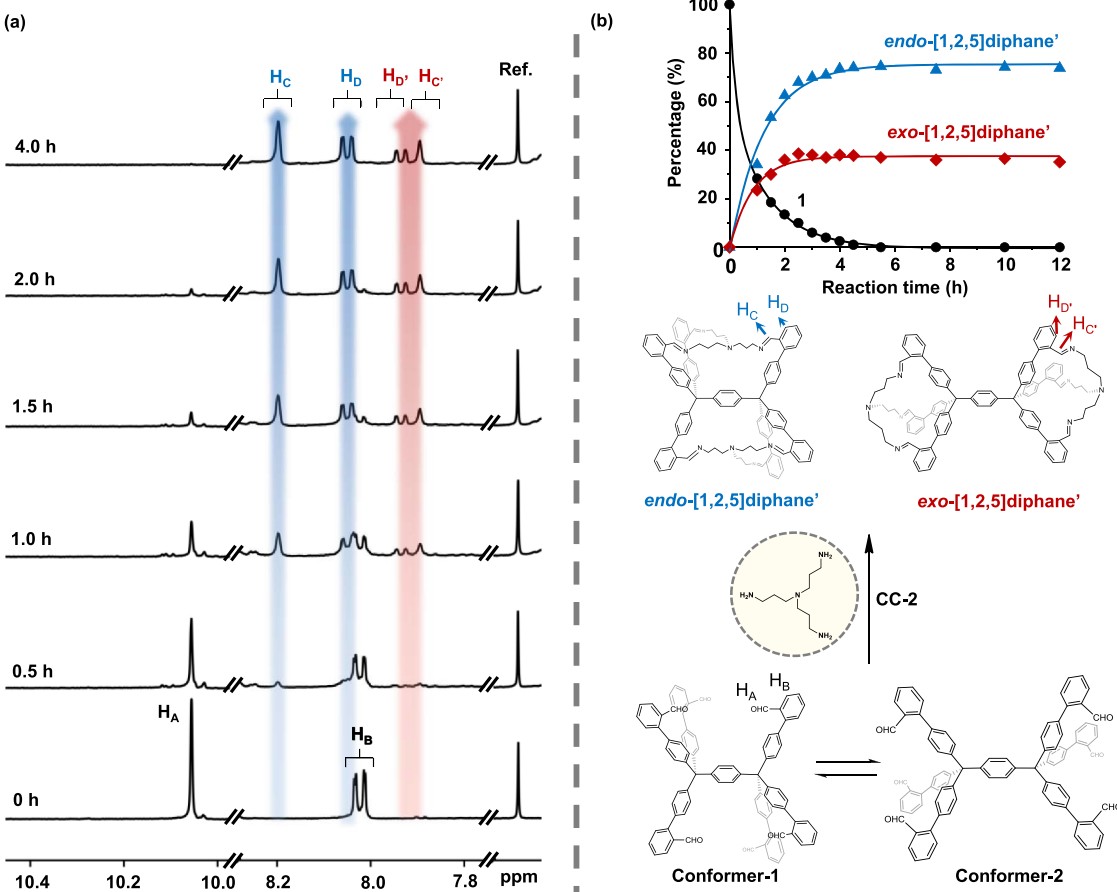

**Fig. 6 $^1$H NMR study of the conformer capturing the progress of Conformer-1 and -2 with large-sized CC-2 as a function of time. a** The $^1$H NMR spectra indicate that Conformer-1 and -2 are captured simultaneously with CC-2. **b** The evolution profile of molecule **1** in gray, *endo*-[1,2,5]diphane' in blue and *exo*-[1,2,5]diphane' in red, determined by the ratios of integral areas of the corresponding characteristic peaks with the internal reference peak of dioxane. $H_A$ represents the proton of aldehyde group of molecule **1**, $H_B$ the nearest proton to $H_A$; $H_C$ the proton of schiff base of *endo*-[1,2,5]diphane', $H_D$ the nearest proton to $H_C$ on the rigid part of the molecule; $H_{C'}$ and $H_{D'}$ the corresponding protons of *exo*-[1,2,5]diphane'.

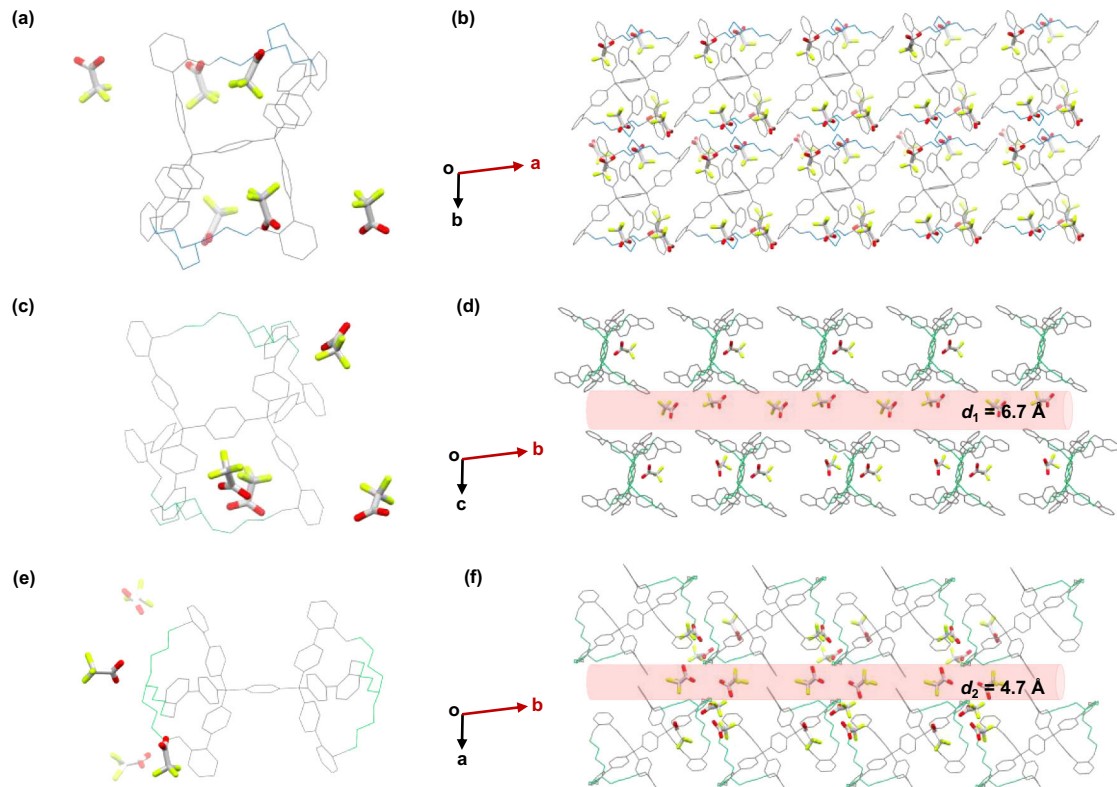

**Fig. 7 The comparison of the molecular structure of three diphanes with the presence of TFA, their self-assembled superstructures with different proton-conducting channels.** Molecular structures of **a** *endo*-[1,2,4]diphane, **c** *endo*-[1,2,5]diphane, and **e** *exo*-[1,2,5]diphane determined by SC-XRD, and their corresponding superstructures shown in **b**, **d**, and **f**. **b** Trifluoroacetic acid (TFA) molecules are trapped within the windows of *endo*-[1,2,4]diphane, whose superstructures exhibit no intermolecular channels. The intermolecular channels highlighted in red are present in the superstructures formed by **d** *endo*-[1,2,5]diphane and **f** *exo*-[1,2,5]diphane, respectively, providing the mobility of TFA molecules. TFA molecules are highlighted with bold stick model, with fluorine atoms labeled in green, oxygen in red, and carbon in gray.

**Table 1 The proton conductivity ($\sigma$) of diphanes–TFA at 303 and 333 K, and their corresponding activation energy ($E_a$) at a relative humidity of 48%.**

| Diphane–TFA | $\sigma$ (S cm$^{-1}$) | | $E_a$ (eV) |
|---|---|---|---|
| | 303 K | 333 K | |
| *endo*-[1,2,4]diphane–TFA | $1.50 \times 10^{-9}$ | $2.44 \times 10^{-9}$ | 0.14 |
| *endo*-[1,2,5]diphane–TFA | $3.17 \times 10^{-6}$ | $1.37 \times 10^{-5}$ | 0.39 |
| *exo*-[1,2,5]diphane–TFA | $1.00 \times 10^{-8}$ | $1.72 \times 10^{-8}$ | 0.15 |

supramolecular synthons for the construction of hierarchical superstructures[36–38] with tuanble properties. For example, here we show that self-assembly of diphanes results in superstructures with tunable proton conductivity. Structural differences on the diphanes structure lead to different molecular packing and distinguished extrinsic channels of the corresponding superstructures. As a result of the structural changes, the conduction efficiency is enhanced up to 1000-fold. This structure–property correlation might provide a reasonable design principle for proton-conducting materials. Other potential applications of diphane-based materials related to ferroelectricity, selective separation and catalysis, are currently ongoing in our laboratories. Finally, the self-assembly approach reported herein is also suitable for preparing the cage-like compounds with two different pockets, denoted Janus diphanes, which we will report in due course.

## Methods

**General information.** Dimethyl terephthalate, tetrakis(triphenylphosphine)palladium, phenyl lithium, Scandium(III) trifluoromethanesulfonate, tris(2-aminoethyl)amine, aniline are purchased from Beijing J&K Chemical Co. Ltd. 2-Formylbenzeneboronic acid and tris(3-aminopropyl)amine are purchased from Beijing InnoChem Co. Ltd. All other reagents are bought from commercial sources and used without any purification unless stated. Tetrahydrofuran is dried over sodium/benzophenone under a nitrogen atmosphere before use.

**Synthesis of model compounds.** Model compounds **1** and **2** are sythesized by Pd(PPh₃)₄ catalyzed Suzuki–Miyaura cross-coupling reaction with commercially available molecule (2-formyl phenyl)boronic acid as a reactant, detailed procedures are shown in the Supplementary Information.

**Formation of diphanes**

*endo*-[1,2,4]diphane. Into a 1000 mL flask is charged with 0.005 mol/L model molecule **1** (1.0 g, 0.84 mmol, 1 equiv.) 170 mL in CHCl₃, then a 0.005 mol/L solution of tris(2-aminoethyl)amine (CC-1) (247 mg, 1.7 mmol, 2 equiv.) 500 mL in CHCl₃ is added dropwise followed by addition of Sc(OTf)₃ (248 mg, 0.5 mmol, 0.6 equiv.). The reaction mixture is stirred at room temperature for 4 h. Then, the product is reduced by NaBH(OAc)₃ (2.6 g, 12.6 mmol, 15 equiv.) overnight, the excess reductant NaBH(OAc)₃ is filtered off under vacuum, and the solution is quenched with NaOH solution (2 M, 200 mL), extracted with CHCl₃ (3 × 200 mL), dried over anhydrous Na₂SO₄, and concentrated to give the crude product. Purification by flash column chromatography (DCM/MeOH/NH₃(aq) = 50:2:3, v/v/v) affords the *endo*-[1,2,4]diphane (736 mg, 64%) as a white solid. NMR spectroscopy is conducted with additional TFA for better solubility in MeOD. ¹H NMR (400 MHz, MeOD): δ 7.19–7.60 (m, 44H), 4.17 (s, 4H), 4.12 (d, $J = 8.0$ Hz, 4H), 4.03 (d, $J = 8.0$ Hz, 4H), 3.06–3.10 (m, 4H), 2.91–2.95 (m, 8H), 2.78–2.82 (m, 4H), 2.61–2.69 (m, 4H), 2.46–2.54 (m, 4H); ¹³C NMR (101 MHz, CDCl₃): δ 161.34, 160.95, 160.59, 160.17, 159.69, 159.27, 158.85, 158.44, 147.14, 146.78, 146.51, 144.10, 143.46, 139.72, 139.61, 132.68, 132.20, 132.16, 131.80, 131.47, 131.31, 131.17, 130.90, 130.67, 130.37, 130.15, 129.96, 129.85, 129.70, 129.63, 120.29, 118.34, 117.46, 115.49, 114.64, 112.61, 111.81, 65.76, 55.13, 54.90, 54.68, 54.45,

54.22, 53.99, 53.77, 50.02, 46.62, 45.23; IR (KBr): 3356.8, 3195.3, 3024.3, 2920.4, 2849.9, 2380.9, 1927.9, 1659.8, 1632.5, 1481.3, 1202.1, 1125.1, 1006.5, 823.3, 763.1, 721.6, 582.2 cm$^{-1}$; MALDI-TOF ($m/z$): $(M + H)^+$ calcd. for $C_{98}H_{95}N_8$, 1384.7708; Found, 1384.7883.

*endo-* and *exo-[1,2,5]diphane.* Into a 1000 mL flask is charged with 0.005 mol/L model molecule 1 (1.0 g, 0.84 mmol, 1 equiv.) 170 mL in CHCl$_3$, then a 0.005 mol/L solution of tris(3-aminopropyl)amine (CC-2) (319 mg, 1.7 mmol, 2 equiv.) 500 mL in CHCl$_3$ is added dropwise followed by addition of Sc(OTf)$_3$ (248 mg, 0.5 mmol, 0.6 equiv.) directly. The reaction mixture is stirred at room temperature for 4 h. Then, the product is reduced by NaBH(OAc)$_3$ (2.6 g, 12.6 mmol, 15 equiv.) overnight, the excess reductant NaBH(OAc)$_3$ is filtered off under vacuum and the solution is quenched with NaOH solution (2 M, 200 mL), extracted with CHCl$_3$ (3 × 200 mL), dried over anhydrous Na$_2$SO$_4$, and concentrated to give the crude product. Purification by flash column chromatography (DCM/MeOH/NH$_3$(aq) = 50:2:3, v/v/v) affords the *endo-*[1,2,5]diphane (720 mg, 59%) and *exo-*[1,2,5] diphane (200 mg, 16%) as white solids. Similarly, NMR spectroscopy of *endo-* [1,2,5]diphane is conducted with additional TFA for better solubility in MeOD.

The structural characterization of *endo-*[1,2,5]diphane is as follows: $^1$H NMR (400 MHz, MeOD): δ 7.64–7.32 (m, 52H), 4.42 (s, 4H), 4.37 (m, 12H), 3.02 (br, 8H), 2.85–2.81 (m, 8H), 2.65 (br, 8H), 1.97 (br, 8H), 1.76 (br, 4H); $^{13}$C NMR (101 MHz, CDCl$_3$): δ 162.25, 161.88, 161.51, 161.14, 147.28, 145.47, 144.37, 144.27, 139.99, 139.57, 132.84, 132.77, 132.51, 132.05, 131.98, 131.29, 131.00, 130.32, 130.23, 129.89, 129.75, 129.52, 129.42, 121.72, 118.83, 117.45, 115.95, 114.63, 113.07, 65.51, 44.37, 22.48, 19.56. IR (KBr): 3357.8, 2920.9, 2850.6, 2381.3, 1659.4, 1633.0, 1458.4, 1377.4, 1265.6, 1193.1, 1114.0, 1005.7, 826.5, 759.7, 729.4, 578.8 cm$^{-1}$; MALDI-TOF ($m/z$): $(M + H)^+$ calcd. for $C_{104}H_{107}N_8{}^+$, 1468.8647; Found, 1468.8496.

The structural characterization of *exo-*[1,2,5]diphane is as follows: $^1$H NMR (400 MHz, MeOD): δ 7.63–7.37 (m, 54H), 4.17 (s, 12H), 2.87–2.91 (t, $J$ = 16.0, 8.0 Hz, 12H), 2.33–2.37 (t, $J$ = 16.0, 8.0 Hz, 12H), 1.67 (br, 12H); $^{13}$C NMR (101 MHz, MeOD): δ 163.16, 162.81, 147.78, 144.89, 143.98, 139.56, 132.77, 132.01, 131.76, 131.48, 130.91, 130.67, 129.83, 65.47, 53.31, 46.04, 25.08; IR (KBr): 3356.6, 2954.3, 2922.5, 2851.2, 2322.0, 1673.7, 1460.9, 1377.9, 1201.4, 1019.7, 823.1, 763.9 cm$^{-1}$; MALDI-TOF ($m/z$): $(M + H)^+$ calcd. for $C_{104}H_{107}N_8{}^+$, 1468.8647; Found, 1468.8496.

**Single-crystal X-ray diffraction.** Crystal data for Molecule 1 is collected on a "Bruker APEX-II CCD" diffractometer (Ga–Kα radiation, $\lambda$ = 1.34139 Å, photon II detector). Crystal data for *endo-*[1,2,4]diphane is collected on a "Bruker D8 VENTURE" diffractometer (Cu–Kα radiation, $\lambda$ = 1.54178 Å, photon II detector). Crystal data for *endo-*[1,2,5]diphane is collected on a "Bruker APEX-II CCD" diffractometer (Ga–Kα radiation, $\lambda$ = 1.34139 Å, photon II detector). Crystal data for *exo-*[1,2,5]diphane is collected on a "Bruker APEX-II CCD" diffractometer (Cu–Kα radiation, $\lambda$ = 1.54178 Å, photon II detector). Crystal data for Cage-2 is collected on a "Bruker D8 VENTURE" diffractometer (Ga–Kα radiation, $\lambda$ = 1.34139 Å, photon II detector).

**Powder X-ray diffraction.** Powder X-ray diffraction (PXRD) patterns of diphanes are recorded on a Bruker D8 advance diffractometer with Cu–Kα1 radiation ($\lambda$ = 1.5406 Å).

**Impedance spectroscopy.** Impedance spectroscopies are collected on a Biologic SP–300 potentiostat using a sinusoidal perturbation of 100 mV over the frequency range 100–7 MHz.

## Data availability
The X-ray crystallographic data (cif and checkcif files) for the structures reported in this study have been deposited at the Cambridge Crystallographic Data Centre (CCDC), under deposition numbers 2033542–2033546. These data files can be obtained free of charge via www.ccdc.cam.ac.uk/data_request/cif. The data supporting the findings of this study are available within the article, its Supplementary Information, or from the corresponding authors upon reasonable request.

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

## Acknowledgements

This work was supported by National Natural Science Foundation of China (21890733, and 22071153), and the Shanghai Natural Science Foundation (18ZR1420800). We thank Dr. Hang Wang (MALDI-TOF MS), Ling-Ling Li (SC-XRD), Qunli Rao and Ningjin Zhang (PXRD), Bona Dai (ssNMR) at the Instrumental Analysis Centre of SJTU. We thank Dr. Shaohui Lin and Dr. Liang Zhang for constructive discussion and suggestions.

## Author contributions

Z.Y. and S.Z. conceived and designed this work. Z.Y., P.L., J.C., K.J.W. Q.-Y.Z. and X.L. performed the experiments. Z.Y. and S.Z. wrote the manuscript. C.Y. contributed to the theoretical calculations. J.D. and X.Z. contributed to the measurement of proton conductivity. H.L. and Y.Y. conducted the PXRD experiments. L.C., Y.-Q.Z. and S.Z. discussed the results and participated in the preparation of the paper.

## Competing interests

The authors declare no competing interests.
