## [Peer Review File · Nature Communications]

A class of organic cages featuring twin cavitiesREVIEWER COMMENTS

Reviewer #1 (Remarks to the Author):

In the manuscript titled “A Size-Matching Strategy to Differentiate Flexible Conformers for the Discovery of Novel Cages with Twin Cavities” Professor Zhang and co-workers present a strategy by which conformationally flexible conformers of hour-glass-shaped molecule 1 can be distinguished by covalently locking the energetically favored conformer(s) via imine formation between formyl functionalities of 1 and two different tripodal amines. The presented strategy is elegant yet simple and, although imine formation chemistry is used in similar fashion in the many facets of supramolecular chemistry to bring about (the desired) minimum energy structure of a system, to my knowledge no prior work exists where targeted identification of conformers is presented in similar manner. The validity of the applied approach is highlighted by the ^1H NMR spectra of the resulting “diphanes” in which the two different viable conformers are well resolved. Further validation is presented in the form of by X-ray crystallographic and computational investigations. My only minor critique of the presented strategy relates to its restriction by the self-correcting nature of DCC, which limits its applicability to a broader group of potentially interesting conformationally flexible compounds, and perhaps, therefore, attractiveness to even wider audience.

My main criticism is related to some aspects of the X-ray crystallography part of the work. These are discussed in detail below and should be addressed before further considering of the publication of the work.

There are no major problems in the presented single crystal X-ray analyses, which are an important part regarding the main arguments and findings of the work. There are some minor issues that need resolving and commenting; these are listed at the end of this report. There are more significant issues, however, in the powder X-ray diffraction (PXRD) part of the work, which affect, for example, how the structure-property correlation regarding proton conductivities is interpreted.

Firstly, there is a significant lack of details in the description of the PXRD sample preparation which makes the reproduction of the work difficult. Are PXRD measurements carried out using the ‘as synthesized’ bulk samples, with or without the addition of TFA, or ground single crystals? These details should be presented in clear fashion in the experimental part.

Secondly, visual comparison between the experimental PXRD patterns and the respective simulated patterns, corresponding to the single crystal X-ray diffraction data, suggests that the experimental and simulated patterns are not similar. If this is indeed the case – i.e. the respective powder samples and the single crystals of endo-[1,2,4]diphane, endo-[1,2,5]diphane and exo-[1,2,5]diphane are not structurally similar – there is an issue in using the single crystal structures to discuss the solid state molecular organization of the diphanes in the context of proton conductivity, assuming that the proton conductivity pellet samples were made using the bulk samples, not single crystals. In other words, the current interpretation on the correlation between the proton conductivities and differences in molecular packing, including TFA mobility, of the diphanes is not merited without further evidence on the structural similarity of the crystal structures and the proton conductivity samples. To clarify:

- Details of the PXRD sample preparation should be included in experimental section.
- The PXRD patterns that are presented should be of the same bulk material used in the preparation of the proton conductivity samples.
- The simulated and experimental PXRD patterns should be analyzed in terms of their similarity, preferably by Rietveld or Pawley fitting.
- If there is no evidence on structural correlation between the experimental and simulated patterns, the use of single crystals structures to draw conclusions on the structural properties of the bulk, including the possible TFA transportation in the channels of the solid material, is not warranted.

I would also like to ask the authors to provide further proof on the claim that transportation of TFA molecules is a significant factor in the different proton conductivities of the synthesized diphanes. In

this context, I suggest reexamining the methodological part as well as the discussion in the paper Nat. Commun. 7, 12750 (2016), where the authors describe only the disordered chloride anions and water molecules to be mobile. None of the TFA anions show any signs of disorder in the crystal structures reported in the present work.

List of specific comments regarding X-ray analyses that need resolving:

- N–H proton missing from atom N48_1 in the structure refinement of exo-[1,2,5]diphane
- Typographical error in cif of endo-[1,2,4]diphane describing the color of the crystal
- Incorrect and missing parameters (absorption correction and moiety formula) in cif file of exo-[1,2,5]diphane. Also, many of the parameters in Table do not correspond to those reported in the cif. Please check and change accordingly.
- Lacking or incorrect single crystal X-ray instrument details in cif of exo-[1,2,5]diphane the device type is assigned as Bruker Apex-II, whereas in the ESI Bruker D8 Venture is given.
- Single crystal X-ray instrumentation should be described in more detail. The text should give the full details of the (two?) instrument(s) with used X-ray source(s) (Cu, Ga?) as well as the detector(s).
- Each of the crystal structures seems to contain voids in the crystal lattice with unresolved electron density. This should be mentioned/discussed and details of the Olex2 Mask/Platon SQUEEZE procedure, which has been used to treat this, should be given in the supplementary information.
- According to the X-ray analyses, the strained endo-[1,2,4]diphane is the only diphane structure which does not seem to have the tertiary amines protonated, perhaps due to the steric shielding of the amine moiety caused by rigid strained conformation. I believe this is an interesting point and worth discussing in the text.

General remarks:

- Spelling mistake in Abstract, line 2 “fileds” -> “fields”
- Spelling mistake on page 4, in the beginning of Results: “As briefly mentioned above, the geometry optimizations (DFT, B3LYP/6-31G*) showed that molecule...”
- In page 4, third row from the bottom, “spectroscopies” should be changed to “spectroscopy”
- In page 4, sentence “Its single crystals suitable for X-ray crystallography were obtained by slow evaporation of solvent from its chloroform solution, but only Conformer-1 could be identified” would benefit from rephrasing. I believe that the authors wish to say that in the crystal structure of 1 all molecules adopt the conformation corresponding to Conformer-1. However, the sentence can also be understood that both conformers existed in the crystal structure, but only Conformer-1 could be identified. Therefore, please clarify the sentence.
- In page 6, please rephrase the sentence “The single crystals suitable for X-ray analysis were obtained by slow evaporation of its THF solution with additional trifluoroacetic acid (TFA) for better solubility. It crystallized into triclinic...”, in order make clear to the reader what “its” and “it” refer to
- In page 7, please change “The twisting of these three chains are...” to “The twisting of these three chains is...”
- Spelling mistake in caption of Fig. 4, “trancated” should read “truncated”
- In page 13, please add “as” to the sentence “These molecules with unique shapes and configurations can also be as used novel supramolecular synthons...”

Reviewer #2 (Remarks to the Author):

The paper by Zhenyu Yang et al. describes the reaction of a hexaaldehyde substrate with two triamines to form various imine double-cages. The hexaaldehyde is conformationally semi-labile – it is partially rigidified by biaryl-type of connection of aromatic rings but it retains low-barrier rotations about single bonds. The unique feature of this hexaaldehyde is the possibility of forming two types of double clefts with the 3-fold symmetry that substantially differ by dimensions (among the unlimited number of other conformations). The authors reacted this hexaladehyde with two aliphatic triamines and obtained imines by a reversible type of reaction. The short triamine formed only one type of a double-cage utilizing smaller clefts, while the longer triamine was able to produce two types of double-

cages utilizing either small or larger clefts. For the obtained double-cages, interesting proton transporting properties have been reported, that rely on the high charge and porosity in the solid-state.

Despite the fact that I think that the “dual-purpose” hexaaldehyde is interesting, and resulting double-cages may exhibit unique properties, I can't agree with the “methodological” interpretation of the results that the authors provided. The authors gave a new name (“Conformation capturing”) to the well-known and intuitive phenomenon. It is quite typical that conformationally labile compounds, especially those with low activation energy barriers can easily adjust and form the products that thermodynamically most stable. It is also straightforward, that for a “small” partner, that does not fit to the large cleft only one type of product is formed while for the larger but still flexible partner two types of product are possible and formed. The idea of multidimensional reactivity that depends on the conditions (the reaction partner, template, or environment) and relies on size-match and conformational shape-adjustment constitutes the core of dynamic covalent chemistry. Therefore, the results presented in this paper are not “a novel yet simple size-matching strategy to identify and distinguish the interconverting conformational” a new strategy to “capture” conformers or to probe conformational space that “had remained unexplored” as the authors claim, but the typical and intuitive behavior of conformationally labile molecules participating in reversible reactions.

I think that the compounds that the authors synthesized are indeed interesting and create many new possibilities that can, in the future, become suitable for publication in Nat. Commun., but, in this interpretation, the authors missed the point. Instead of focusing on the unique properties of the double cage products, which are way more interesting (but require more work and more examples), they described rather typical and intuitive behavior, presenting it as a new methodology. Therefore, I recommend the rejection of the paper in the current form. The results may become suitable for publication in the future, with the focal point changed and with more results about the obtained double cages.

Additional remarks:

PXRD patterns – experimental and simulated are very different. Most likely more polymorphic forms are present in the samples.

It is quite typical that for the conformationally labile compound only one of the possible conformers is found in the crystal phase. So depriving the value of X-ray analysis by saying “SC-XRD, only able to identify Conformer-1 in solid-state” is not OK.

The discussion part does not contain a discussion but a summary- so either some discussion should be added or the part should be called summary.

The authors should clearly state either the calculation/conclusions/discussion is for the imine or the amine in the protonated or non-protonated form because the torsional requirements and thus energies can be very different.

HPLC analysis of the distribution of products (after quenching) would be informative

The part starting from “No interchange between these two imine-based isomers has been observed throughout their formation process” and the conclusion that follows are not proved. The net result of the equilibrium does not change, but there are no proofs that the exchange processes are locked.

Reviewer #3 (Remarks to the Author):

See the attached file [included below]

Reviewer #3 Attachment:

In this manuscript, Z. Yang et al. provides a novel strategy to differentiate and identify rapid interconverting conformational isomers which would be otherwise difficult to be detected by standard characterization techniques. The simple but interesting approximation consists of the idea to catch conformational isomers of a targeted system (precursor) by reacting them through dynamic covalent chemistry with molecular units (capturers) of different size. The strategy is demonstrated for a three-dimensional hexaformyl model precursor with an extremely-rich conformational space (*i.e.*, infinite conformers), which can be broadly classified into two distinct type of conformers. The outcome of the reported strategy (final species) not only allows the identification of two elusive conformers of the precursor but also provides new molecular cages with potential functionalities and applications (e.g., favorable proton conduction).

Within my limited knowledge about the literature of molecular cages, the proposed synthetic method (key result of the study) looks quite simple and relatively easy to be used for the design of novel 3D molecular cages. It is difficult, however, to me to assess the significance and broad scope of the proposed approach for a general scientific community in the frontier of chemistry, materials chemistry and chemical biology. Nevertheless, the relevance of the synthetic procedure for the design of novel and exotic molecular cages can be strengthened and highlighted with a more extended introduction (only a few sentences) emphasizing the advantages of this new approach with respect to the previous ones to obtain organic molecular cages.

The study is very well-organized, well-written and can be easily followed with the main idea clear along the manuscript. I have really enjoyed the paper. Nonetheless, a criticism, related with the previous one (significance), would be to provide a sufficient context with more comparisons with previous works to make clear the relevance of the study.

Concerning validity of the study, the experimental findings seem to be solid and support the conclusions drawn. Nevertheless, I believe that the theoretical part, which is quite relevant in the study, can be improved. In this respect, I would suggest the following points to be considered:

- 1) The experimental analysis (H NMR) about the conformational interconversion of **1** is performed in solution. However, DFT calculations seem to be performed in gas phase. That effect may have an effect on the energy difference between **conformer-1** and **-2**. Therefore, I would include solvent effects in the calculations (a continuum model can be enough for this study).
- 2) van der Waals (vdW) intramolecular interactions can also have an effect on the energy difference between **conformer-1** and **-2**. Plain B3LYP functional cannot capture vdW interactions and should be augmented with external approximations (the most practical would be the Grimme's D3 approach). I would re-do the calculations at least at the B3LYP-D3/6-31G* level to check that intramolecular vdW interactions are not playing an important effect.
- 3) As the energy difference between **conformer-1** and **-2** is quite small even in the range of error of the DFT method, I would check that the energy order is preserved with another density functional. In this sense, extra calculations with the M06-2X, which can describe

vdW interactions at short range, can be a good alternative to check that the theoretical results are robust.

- 4) In the caption of Fig. 2a, the authors write: “Fast interconversion of **Conformer-1** and **2** with extremely low isomerization barrier.” However, the authors have not computed the isomerization barrier, they provide an energy difference between both conformers. An estimate of the low isomerization barriers can be provided by calculating the torsional potential through: i) the rotation of the carbon-carbon bond of the central benzene ring and the subsequent sp³ carbon atom and ii) the rotation around the inter-ring carbon-carbon bond of the two aryl rings within a peripheral arm.

Then the author says: “Free energy of each isomer was determined by geometry optimization (DFT, B3LYP/6-31G*)”. I was wondering if the energy provided is actually a free energy difference computed from vibrational calculations or is simply the energy difference calculated between the optimized **conformer 1** and **2**. If it is obtained by the latter approach is not a free energy estimate.

- 5) In the Supplementary Information (DFT calculations section), the author defines the strain energy as the energy difference between the EC unit found in the cage and that obtained after optimization. The abbreviation EC is not defined.
- 6) In the Supplementary Information, there is a section devoted to Molecular dynamics simulations. However, the authors do not call to this section during the discussion in the main text which is weird. On the other hand, I do not understand the histograms in Supplementary Figure 4, is it possible that they are interchanged? If it is the case, it would have more sense to me. Nevertheless, I think that Supplementary Figure 4 deserves a brief discussion at least in the Supplementary Information.
- 7) Apart from the strain energies evaluated for the four diphane cages, the relative difference between the two types of cages (*exo* and *endo*) should be provided to see if they can correlate with the experimental yield obtained for the final cages.

Typo in the caption of Fig. 4 (second line). “truncated” should be substituted by “truncated”

As a future suggestion (not for this paper), it would be great if you are able to have control and break the final molecular cage by external stimuli but isolating the conformational isomers.

Reviewer 1:

In the manuscript titled “A Size-Matching Strategy to Differentiate Flexible Conformers for the Discovery of Novel Cages with Twin Cavities” Professor Zhang and co-workers present a strategy by which conformationally flexible conformers of hour-glass-shaped molecule **1** can be distinguished by covalently locking the energetically favored conformer(s) via imine formation between formyl functionalities of **1** and two different tripodal amines. The presented strategy is elegant yet simple and, although imine formation chemistry is used in similar fashion in the many facets of supramolecular chemistry to bring about (the desired) minimum energy structure of a system, to my knowledge no prior work exists where targeted identification of conformers is presented in similar manner. The validity of the applied approach is highlighted by the ^1H NMR spectra of the resulting “diphanes” in which the two different viable conformers are well resolved. Further validation is presented in the form of by X-ray crystallographic and computational investigations. My only minor critique of the presented strategy relates to its restriction by the self-correcting nature of DCC, which limits its applicability to a broader group of potentially interesting conformationally flexible compounds, and perhaps, therefore, attractiveness to even wider audience.

Answer: First of all, we would like to thank the Reviewer for carefully reviewing our manuscript and appreciating our research work. As mentioned by the Reviewer, self-correcting DCC is a useful strategy for the preparation of organic cages, which we fully agree. On the other hand, we rather use DCC as a synthetic tool to discover interesting structures related to the flexible conformations of a precursor, including but not limited to the three *diphanes* reported in the present work.

For example, by employing the size-matching strategy, the precursor molecule **2** reacted with large-sized **CC-2** to selectively yield monofunctional **Cage-2** with a single cavity and one pendant -OH (Fig. C1a), which is presented as a by-product in the original manuscript (Fig. 5 and Fig. C1a); on the other hand, the reaction between molecule **2** and small-sized **CC-1** selectively produced bifunctional **Cage-3** with two -OH at both ends of the molecule (Fig. C1a). We are currently polymerizing these two cages (as monomers) to form lateral and main-chain polycages, respectively, which will be reported in due course. Besides, we found that **Cage-2** (not **Cage-1**) self-assembled into superstructures with distinct four

hierarchical levels (Fig. C2). The tertiary structure is supramolecular 2₁-helix, and an array of these helices formed the quaternary superstructure.

Figure C1. Schematic representation of the synthesis procedure. (a) Selective synthesis of **Cage-2** and **Cage-3** with mono- and bifunctional groups *via* size-matching strategy. Large-sized tris(3-aminopropyl)amine (**CC-2**) molecule yields **Cage-2** as major product and trace amount of **Cage-4** *via* dynamic reversible imination reaction and subsequent reduction, while small-sized tris(2-aminoethyl)amine (**CC-1**) molecule produces **Cage-3** as the major product and **Cage-1** is the by-product (trace amount). (b) The synthesis of **R-endo-[1,2,4]diphane** and **S-endo-[1,2,4]diphane** *via* dynamic reversible imination reaction and subsequent reduction.

Figure C2. Hierarchical self-assembly of **Cage-2** during crystallization. a) Side view of its crystal structure (primary structure), b) supramolecular tetramer (secondary structure) self-assembled by four **Cage-2** molecules, c) supramolecular 2_1 -Helix self-organized by supramolecular tetramer, and d) the super lattice (quaternary structure) formed by an array of 2_1 -Helices. Hydrogen atoms and solvents are omitted for clarity, and the crystal belongs to orthorhombic space group $P2_1/n$.

The conformation tethering strategy also allowed us to prepare a series of chiral *diphanes* such as *R-endo*-[1,2,4]diphane, *S-endo*-[1,2,4]diphane, *R-endo*-[1,2,5]diphane and *S-endo*-[1,2,5]diphane (Fig. C1b). Taking *R-endo*-[1,2,4]diphane for instance, it also self-assembled into superstructures with four hierarchical levels (Fig. C3). It is worth noting that chiral channels were formed in the tertiary structures, which allows us to explore the induced chirality and spontaneous polarization of confined water. It endows the superstructure with interesting properties such as *non-linear optics* and *ferroelectrics*, which are under investigation in our lab.

In a word, the strategy reported in the current study provides a series of interesting molecular platform for the exploration of novel hierarchical superstructures, which allows us to search for emergent properties of these superstructures.

Figure C3. Hierarchical self-assembly of *R-endo*-[1,2,4]diphane during crystallization. a) Side view of its crystal structure (primary structure), b) supramolecular dimer (secondary structure) self-assembled by two *R-endo*-[1,2,4]diphane molecules, c) supramolecular 3_2 -Helix self-organized by supramolecular dimer, forming a chiral intermolecular channel as shown in blue, and d) the super lattice (quaternary structure) formed by an array of 3_2 -Helices. Hydrogen atoms and solvents are omitted for clarity, and the crystal belongs to trigonal space group $P3_221$.

➤ In the powder X-ray diffraction (PXRD) part of the work.

1. Firstly, there is a significant lack of details in the description of the PXRD sample preparation which makes the reproduction of the work difficult. Are PXRD measurements carried out using the ‘as synthesized’ bulk samples, with or without the addition of TFA, or ground single crystals? These details should be presented in clear fashion in the experimental part.

Answer: Thanks for the reviewer’s comment. The procedure of sample preparation for PXRD measurements was referred to the paper *Nat. Commun.* **7**, 12750 (2016), and its detailed information is as follows: “The as-synthesized bulk powder samples of *diphanes* were firstly dispersed in distilled THF with a concentration of 30 mg/mL, followed by sonication until thorough dispersion. Additional TFA was then added until the mixture became clear, which was subsequently subjected to filtration with 0.45 μm PTFE spring filter to remove the undissolved residue. Slow evaporation of the settled solution yielded large quantity of high-quality single crystals, which were then loaded into a 0.7 mm diameter quartz glass capillary. PXRD data were collected at ambient temperature with a capillary

setup". It has been added and highlighted in yellow in the Techniques part of the Revised Supplementary Information.

2. Secondly, visual comparison between the experimental PXRD patterns and the respective simulated patterns, corresponding to the single crystal X-ray diffraction data, suggests that the experimental and simulated patterns are not similar. If this is indeed the case – i.e. the respective powder samples and the single crystals of *endo*-[1,2,4]diphane, *endo*-[1,2,5]diphane and *exo*-[1,2,5]diphane are not structurally similar – there is an issue in using the single crystal structures to discuss the solid state molecular organization of the diphanes in the context of proton conductivity, assuming that the proton conductivity pellet samples were made using the bulk samples, not single crystals. In other words, the current interpretation on the correlation between the proton conductivities and differences in molecular packing, including TFA mobility, of the diphanes is not merited without further evidence on the structural similarity of the crystal structures and the proton conductivity samples. To clarify:

Q1: Details of the PXRD sample preparation should be included in experimental section.

A1: The detailed procedure of sample preparation for PXRD measurement has been added in the Techniques part of the Revised Supplementary Information.

Q2: The PXRD patterns that are presented should be of the same bulk material used in the preparation of the proton conductivity samples.

Q3: The simulated and experimental PXRD patterns should be analyzed in terms of their similarity, preferably by Rietveld or Pawley fitting.

Q4: If there is no evidence on structural correlation between the experimental and simulated patterns, the use of single crystals structures to draw conclusions on the structural properties of the bulk, including the possible TFA transportation in the channels of the solid material, is not warranted.

A2-4: Thanks for the reviewer's comments, which are indeed correct and tremendously help us improve the rigorousness of our work. According to the reviewer's suggestion, we reinvestigated the correlation between the molecular packing in bulk samples and the proton conductivities of the three *diphanes*. These three questions can be answered together as follows.

- 1) Regarding the consistency between PXRD and SC-XRD, we prepared the bulk materials from large quantity of single crystals of each *diphane*, compared their PXRD patterns (both experimental and Pawley refinement) with the pattern simulated from SC-XRD data (Supplementary Fig. 13-15), and used the same samples to conduct the experiments of proton

conductivity.

Supplementary Figure 13. Powder X-ray pattern of *endo*-[1,2,4]diphane, experimental pattern (*grey line*), pawley refinement (*blue circles*), simulated pattern (*red line*) and difference (*black line*) profiles for Pawley refinement ($R_{wp} = 3.56\%$, $R_p = 1.56\%$) at 298 K ($a = 13.717759$, $b = 15.453886$, $c = 16.448906$ Å, $P-1$). The reflection positions are marked with green.

Supplementary Figure 14. Powder X-ray pattern of *endo*-[1,2,5]diphane, experimental pattern (*grey line*), pawley refinement (*blue circles*), simulated pattern (*red line*) and difference (*black line*) profiles for Pawley refinement ($R_{wp} = 3.31\%$, $R_p = 3.93\%$) at 298 K ($a = 16.421116$, $b = 16.786190$, $c = 19.463487$ Å, $P-1$). The reflection positions are marked with green.

Supplementary Figure 15. Powder X-ray pattern of *exo*-[1,2,5]diphane, experimental pattern (*grey line*), pawley refinement (*blue circles*), simulated pattern (*red line*) and difference (*black line*) profiles for Pawley refinement ($R_{wp} = 3.36\%$, $R_p = 1.10\%$) at 298 K ($a = 16.303037$, $b = 16.787783$, $c = 19.356683$ Å, $P-1$). The reflection positions are marked with green.

As can be seen from Supplementary Fig. 13-15, the PXRD pattern of each sample fits well with the pattern simulated from SC-XRD data, with R_{wp} and R_p below 5%. These figures were newly collected, which have been amended and highlighted in yellow in the Revised Supplementary Information.

- 2) As the structural analysis of each bulk material was ascertained, we set out to revise the correlation between molecular packing and the proton conductivity of the three *diphanes* (Supplementary Table 7 and Fig. 18-21). The results showed that *endo*-[1,2,5]diphanes packed into porous crystals with the largest and ordered channel, *exo*-[1,2,5]diphanes with zig-zag and narrower channel, while densely packed *endo*-[1,2,4]diphanes with no obvious channels, and the proton conductivity agreed well with the order *endo*-[1,2,5]diphanes > *exo*-[1,2,5]diphanes > *endo*-[1,2,4]diphanes (Supplementary Table 7 and Fig. 18-21) **This tendency is identical in the original results.** On the other hand, the activation energies of *diphanes* are below 0.4 eV, which are lower than the original results that are no less than 0.79 eV. The revised results therefore indicate their proton conductivity follows Grotthuss mechanism, which involves in proton hopping from one stationary oxygen atom to an adjacent oxygen atom. The investigation of the role of TFA and the mechanism will be provided in the next Q&A.

Supplementary Table 7. Proton conductivities at different temperatures under the air humidity

(48% RH).

		Temperature (°C)	30	40	50	60
endo -[1,2,4]diphane	R (Ω)		2.01×10^7	1.67×10^7	1.43×10^7	1.23×10^7
	σ ($S\ cm^{-1}$)		1.50×10^{-9}	1.80×10^{-9}	2.11×10^{-9}	2.44×10^{-9}
endo -[1,2,5]diphane	R (Ω)		8.82×10^3	5.27×10^3	4.27×10^3	2.19×10^3
	σ ($S\ cm^{-1}$)		3.17×10^{-6}	5.71×10^{-6}	7.04×10^{-6}	1.37×10^{-5}
exo -[1,2,5]diphane	R (Ω)		3.00×10^6	2.40×10^6	2.19×10^6	1.75×10^6
	σ ($S\ cm^{-1}$)		1.00×10^{-8}	1.25×10^{-8}	1.37×10^{-8}	1.72×10^{-8}

Supplementary Figure 18. Nyquist plots of *endo*-[1,2,4]diphane at different temperatures with the air humidity of 48% RH.

Supplementary Figure 19. Nyquist plots of *endo*-[1,2,5]dipane at different temperatures with the air humidity of 48% RH.

Supplementary Figure 20. Nyquist plots of *exo*-[1,2,5]dipane at different temperatures with the air humidity of 48% RH.

Supplementary Figure 21. Arrhenius plots of *diphanes* and their corresponding activation energy. Temperature data were collected at the air humidity of 48% over the temperature range 303.15-333.15 K.

3 I would also like to ask the authors to provide further proof on the claim that transportation of TFA molecules is a significant factor in the different proton conductivities of the synthesized *diphanes*. In this context, I suggest reexamining the methodological part as well as the discussion in the paper *Nat. Commun.* 7, 12750 (2016), where the authors describe only the disordered chloride anions and water molecules to be mobile. None of the TFA anions show any signs of disorder in the crystal structures reported in the present work.

Answer: Thanks for the reviewer's comment. In order to carefully investigate the proton conductivity mechanism including the motion of TFA molecules during the proton conduction, we first measured the proton conductivity of *diphanes* without adding TFA molecules to verify its necessity for proton conductivity (Supplementary Fig. 17). The results showed that the *diphanes* had high impedance values without adding TFA molecules, **which indicates that doping the *diphanes* with TFA could significantly enhance their proton conductivity by providing an excess of protons.**

Supplementary Figure 17. Nyquist plots of (a) *endo*-[1,2,4]diphane, (b) *endo*-[1,2,5]diphane and (c) *exo*-[1,2,5]diphane without doping of TFA.

Secondly, we investigated the dynamics of *diphanes-TFA* (acid doped form of the powder samples used for PXRD) by using various solid-state NMR (ssNMR) technique, which is commonly used to investigate the mechanism of proton conductivity (ref. 32-34 in the Revised Manuscript). According to the comparison of the single-pulse ^1H MAS and dipolar-based 1D ^1H DQ/SQ NMR spectra (Supplementary Fig. 22-24), it was found that the N-H resonances of *diphanes*, and their protonic peaks containing excess TFA in the channels of *diphanes-TFA* were almost suppressed, while the rigid protons remained in 1D ^1H DQ/SQ NMR spectra. These results clearly indicate that the N-H related protons of *diphanes*, together with the excess TFA molecules in the channels of *diphanes-TFA*, are highly mobile, whose dipolar interactions are remarkably reduced by molecular motions. Besides, the conductivities of *diphanes* were significantly enhanced by doping with additional TFA, where the increased number of proton carriers are essential for the formation of excess protons in the hydrogen bond network, which facilitates the proton transportation.

Supplementary Figure 22. (a) 1D ^1H single-pulse spectrum of *endo*-[1,2,4]diphane. (b) The comparison of ^1H single-pulse spectra between *endo*-[1,2,4]diphane (black line) and *endo*-[1,2,4]diphane-TFA (red line) (c) 1D ^1H single-pulse (black line) and ^1H DQ/SQ spectra (blue line) for *endo*-[1,2,4]diphane. (d) ^1H single-pulse (black line) and ^1H DQ/SQ spectra (green line) for *endo*-[1,2,4]diphane-TFA. *endo*-[1,2,4]Diphane-TFA is protonated crystal powder of *endo*-[1,2,4]diphane, which are formed by slow evaporation of its THF solution with additional TFA.

Supplementary Figure 23. (a) 1D ^1H single-pulse spectrum of *endo*-[1,2,5]diphane. (b) The comparison of ^1H single-pulse spectra between *endo*-[1,2,5]diphane (black line) and *endo*-[1,2,5]diphane-TFA (red line) (c) 1D ^1H single-pulse (black line) and ^1H DQ/SQ spectra (blue line) for *endo*-[1,2,5]diphane. (d) ^1H single-pulse (black line) and ^1H DQ/SQ spectra (green line) for *endo*-[1,2,5]diphane-TFA. *endo*-[1,2,5]Diphane-TFA is the protonated crystal form of *endo*-[1,2,5]diphane, which were formed by slow evaporation of its THF solution with additional TFA.

Supplementary Figure 24. (a) 1D ^1H single-pulse spectrum of *exo*-[1,2,5]*diphane*. (b) The comparison of ^1H single-pulse spectra between *exo*-[1,2,5]*diphane* (black line) and *exo*-[1,2,5]*diphane-TFA* (red line) (c) 1D ^1H single-pulse (black line) and ^1H DQ/SQ spectra (blue line) for *exo*-[1,2,5]*diphane*. (d) ^1H single-pulse (black line) and ^1H DQ/SQ spectra (green line) for *exo*-[1,2,5]*diphane-TFA*. *exo*-[1,2,5]*Diphane-TFA* is protonated crystal powder of *exo*-[1,2,5]*diphane*, which are formed by slow evaporation of its THF solution with additional TFA.

Accordingly, we added a discussion “Although the exact location of water molecules within these superstructures is currently uncertain, their activation energies are lower than 0.4 eV, indicating the proton conduction was realized *via* Grotthuss mechanism, where protons are hopping within the channels through interconnected hydrogen bonding.³⁰⁻³¹ In addition, the dynamics of *diphanes-TFA* were further investigated through various solid-state NMR (ssNMR) spectra.³²⁻³³ It was found from the comparison of the single-pulse ^1H MAS and dipolar-based 1D ^1H DQ/SQ NMR spectra (Supplementary Fig. 22-24) where the N-H resonances of *diphanes*, and their protonic peaks containing excess TFA in the channels of *diphanes-TFA* were almost suppressed, while the rigid protons remained in 1D ^1H DQ/SQ NMR spectra. These results clearly indicate that the N-H related protons of *diphanes*, together with the excess TFA molecules in the channels of *diphanes-TFA* are highly mobile, whose dipolar interactions are remarkably reduced by molecular motions. Furthermore, the conductivities of *diphanes* are significantly enhanced by doping with additional TFA, where the increased number of protons might acting as charge carriers are essential to the formation of excess protons in the hydrogen bond network, facilitating the proton conduction.³⁴”, which is highlighted in yellow in Page 13 in the Revised Manuscript.

- List of specific comments regarding X-ray analyses that need resolving:

Q1: N–H proton missing from atom N48_1 in the structure refinement of *exo*-[1,2,5]diphane.

A1: N-H proton of N48_1 has been revised in the structure refinement of *exo*-[1,2,5]diphane and the revised CIF has been redeposited at CCDC.

Q2: Typographical error in cif of *endo*-[1,2,4]diphane describing the color of the crystal.

A2: The typographical error word “colouless” describing the color of the crystal in CIF of *endo*-[1,2,4]diphane has been revised as “colourless”, and the revised CIF has also been redeposited at CCDC.

Q3: Incorrect and missing parameters (absorption correction and moiety formula) in cif file of *exo*-[1,2,5]diphane. Also, many of the parameters in Table do not correspond to those reported in the cif. Please check and change accordingly.

A3: In supplementary table 2, crystal data for **molecule 1**.

- i. The empirical formula of Molecule **1** “C₉₀H₆₂C₁₁₂O₆” has been revised as “C₉₀H₆₂Cl₁₂O₆”.
- ii. The calculated density “1.334 Mg/cm³” has been revised as “1.334 Mg/m³”.
- iii. The line of “Reflections collected / unique, 23673/7740 [Rint = 0.0509]” has been revised as “Reflections collected, 23673”. After which, a new row was added and its content is “Independent reflections, 7740 [R(int) = 0.0509]”.

In supplementary table 3, crystal data for ***endo*-[1,2,4]diphane**.

- i. The calculated density “1.170 Mg/cm³” has been revised as “1.170 Mg/m³”.
- ii. The line of “Reflections collected/unique, 35362/10326 [Rint = 0.0960]” has been revised as “Reflections collected, 35362”. After which, a new row was added and its content is “Independent reflections, 10326 [R(int) = 0.0960]”.
- iii. The line of “Absorption correction, None” has been revised as “Absorption correction, Multi-scan”.
- iv. The line of “Max and min. transmission, None” has been revised as “Max and min. transmission, 0.7524 and 0.6163”.

In supplementary table 4, crystal data for ***endo*-[1,2,5]diphane**.

- i. The calculated density “0.942 Mg/cm³” has been revised as “0.942 Mg/m³”.
- ii. The line of “Reflections collected/unique, 44705/17594 [Rint = 0.1319]” has been revised as

“Reflections collected, 44705”. After which, a new row was added and its content is “Independent reflections, 17594 [R(int) = 0.1319]”.

iii. The line of “Completeness to theta = 67.684, 98.6%” has been revised as “Completeness to theta = 53.594, 98.6%”.

iv. The line of “Max and min. transmission, 0.7508 and 0.5085” has been revised as “Max and min. transmission, 0.7508 and 0.2458”

v. The line of “Refinement method, Full-matrix-block least-squares on F²” has been revised as “Refinement method, Full-matrix least-squares on F²”.

In supplementary table 5, crystal data for ***exo*-[1,2,5]diphane**.

i. The empirical formula “C₁₈₄H₂₄₀F₂₄N₈O₃₂” has been revised as “C₁₈₄H₂₄₂F₂₄N₈O₃₂”.

ii. The formula weight “3531.83” has been revised as “3533.84”.

iii. The calculated density “1.252 Mg/cm³” has been revised as “1.253 Mg/m³”.

iv. The value of F(000) “1872” has been revised as “1874”.

v. The line of “Reflections collected/unique, 78259/15999 [Rint = 0.0873]” has been revised as “Reflections collected, 78264”. After which, a new row was added and its content is “Independent reflections, 15998 [R(int) = 0.0873]”.

vi. The line of “Completeness to theta = 53.594°, 99.1%” has been revised as “Completeness to theta = 65.534°, 99.0%”

vii. The line of “Absorption correction, None” has been revised as “Absorption correction, Semi-empirical from equivalents”.

viii. The line of “Max and min. transmission, None” has been revised as “Max and min. transmission, 0.7526 and 0.5965”.

ix. The line of “Data/restraints/parameters, 15999/123/1117” has been revised as “Data/restraints/parameters, 15998/264/1081”.

x. The line of “Goodness-of-fit on F², 1.552” has been revised as “Goodness-of-fit on F², 1.453”

xi. The line of “Final R indices [I > 2 sigma (I)], R1 = 0.1644, wR2 = 0.4142” has been revised as “Final R indices [I > 2 sigma (I)], R1 = 0.1459, wR2 = 0.3884”.

xii. The line of “R indices (all data), R1 = 0.2042, wR2 = 0.4501” has been revised as “R indices (all data), R1 = 0.1864, wR2 = 0.4265”.

xiii. The last line, “Largest diff. peak and hole, 1.533 and -0.490 e.A⁻³” has been revised as “Largest

diff. peak and hole, 0.790 and -0.444 e.A⁻³”.

In supplementary table 6, crystal data for **Cage-2**.

i. The calculated density “1.260 Mg/cm³” has been revised as “1.260 Mg/m³”.

ii. The line of “Reflections collected/unique, 79008/19768 [Rint = 0.1267]” has been revised as “Reflections collected, 79008”. After which, a new row was added and its content is “Independent reflections, 19768 [R(int) = 0.1267]”.

Q4: Lacking or incorrect single crystal X-ray instrument details in cif of *exo*-[1,2,5]diphane the device type is assigned as Bruker Apex-II, whereas in the ESI Bruker D8 Venture is given.

A4: Single Crystal X-ray Diffraction data for Molecule **1** was collected on a “Bruker APEX-II CCD” diffractometer (Ga-K α radiation, λ =1.34139 Å, photon II detector).

Single Crystal X-ray Diffraction data for *endo*-[1,2,4]diphane was collected on a “Bruker D8 VENTURE” diffractometer (Cu-K α radiation, λ =1.54178 Å, photon II detector).

Single Crystal X-ray Diffraction data for *endo*-[1,2,5]diphane was collected on a “Bruker APEX-II CCD” diffractometer (Ga-K α radiation, λ =1.34139 Å, photon II detector).

Single Crystal X-ray Diffraction data for *exo*-[1,2,5]diphane was collected on a “Bruker APEX-II CCD” diffractometer (Cu-K α radiation, λ =1.54178 Å, photon II detector).

Single Crystal X-ray Diffraction data for **Cage-2** was collected on a “Bruker D8 VENTURE” diffractometer (Ga-K α radiation, λ =1.34139 Å, photon II detector).

The above information has been revised in the Method part of the Revised Manuscript.

Q5: Single crystal X-ray instrumentation should be described in more detail. The text should give the full details of the (two?) instrument(s) with used X-ray source(s) (Cu, Ga?) as well as the detector(s).

A5: The question about the detail information about single crystal X-ray instrumentation has been addressed in **A4**, which has also been revised in the Method part of the Revised Manuscript.

Q6: Each of the crystal structures seems to contain voids in the crystal lattice with unresolved electron density. This should be mentioned/discussed and details of the Olex2 Mask/Platon SQUEEZE procedure, which has been used to treat this, should be given in the supplementary information.

A6: Some solvent molecules were highly disordered and could not be reasonably located. The voids in the crystal lattice were treated with Platon squeeze program and the details of unresolved electron density have been added in CIF files, which have been redeposited at CCDC as well. The above

information has also been mentioned in the Techniques part of the Revised Supplementary Information.

Q7: According to the X-ray analyses, the strained endo-[1,2,4]diphane is the only diphane structure which does not seem to have the tertiary amines protonated, perhaps due to the steric shielding of the amine moiety caused by rigid strained conformation. I believe this is an interesting point and worth discussing in the text.

A7: Thanks for the reviewer's careful inspection of the protonated crystal structure. In Page 7 of the Revised Manuscript, we added "In addition, the X-ray structure of *endo*-[1,2,4]diphane showed that the tertiary amines on the aliphatic chains were not protonated, even with the presence of TFA. This is presumably due to the steric hindrance of the irregularly stretched amine moiety caused by the rigid strain conformation, as well as the electrostatic repulsion of the neighbouring TFA molecules." We also added Supplementary Fig. 12 to illustrate this interesting point.

➤ General remarks:

- 1) Spelling mistake in Abstract, line 2 "fileds" -> "fields".
- 2) Spelling mistake on page 4, in the beginning of Results: "As briefly mentioned above, the geometry optimizations (DFT, B3LYP/6-31G*) showed that molecule...".
- 3) In page 4, third row from the bottom, "spectroscopies" should be changed to "spectroscopy".
- 4) In page 4, sentence "Its single crystals suitable for X-ray crystallography were obtained by slow evaporation of solvent from its chloroform solution, but only Conformer-1 could be identified" would benefit from rephrasing. I believe that the authors wish to say that in the crystal structure of 1 all molecules adopt the conformation corresponding to Conformer-1. However, the sentence can also be understood that both conformers existed in the crystal structure, but only Conformer-1 could be identified. Therefore, please clarify the sentence.
- 5) In page 6, please rephrase the sentence "The single crystals suitable for X-ray analysis were obtained by slow evaporation of its THF solution with additional trifluoroacetic acid (TFA) for better solubility. It crystallized into triclinic...", in order make clear to the reader what "its" and "it" refer to.
- 6) In page 7, please change "The twisting of these three chains are..." to "The twisting of these three chains is...".

- 7) Spelling mistake in caption of Fig. 4, “trancated” should read “truncated”.
- 8) In page 13, please add “as” to the sentence “These molecules with unique shapes and configurations can also be as used novel supramolecular synthons...”.

Answer: Thanks for the reviewer’s comments.

- 1) In Page 1, the second line of the abstract, the word “fileds” has been revised as “fields”.
- 2) In Page 4, in the beginning of Results, the sentence “As briefly mentioned above, the geometry optimizatoin (DFT, B3LYP/6-31G*) showed molecule **1** exists infinite forms of conformers via all possible C-C single bond rotations.” has been revised as “As briefly mentioned above, the geometry optimizatoin (DFT, B3LYP/6-31G*) showed that molecule **1** exists infinite forms of conformers via all possible C-C single bond rotations” in the Revised Manuscript.
- 3) In Page 5, the fourth line of the second paragraph, the word “spectroscopies” has been revised as “spectroscopy”.
- 4) In Page 5, the sentence “Its single crystals suitable for X-ray crystallography were obtained by slow evaporation of solvent from its chloroform solution, but only Conformer-1 could be identified” has been revised as “Its single crystals suitable for X-ray crystallography were obtained by slow evaporation of solvent from its chloroform solution, but molecule **1** only exists the conformation corresponding to **Conformer-1** in the crystal structure (Fig. 2c), which is common for the conformationally labile compounds that often crystallize into the most efficient molecular packing.²⁶” in the Revised Manuscript.
- 5) In Page 7, the sentence “The single crystals suitable for X-ray analysis were obtained by slow evaporation of its THF solution with additional trifluoroacetic acid (TFA) for better solubility. It crystallized into triclinic...” has been revised as “The single crystals suitable for X-ray analysis were obtained by slow evaporation of *endo*-[1,2,4]diphane solution in THF with additional trifluoroacetic acid (TFA) for better solubility. *endo*-[1,2,4]Diphane crystallized into triclinic...” in the Revised Manuscript.
- 6) In Page 7, the sentence “The twisting of these three chains are...” has been revised as “The twisting of these three chains is...” in the Revised Manuscript.
- 7) In Page 8, in the caption of Fig. 4, the word “trancated” has been revised as “truncated”.

- 8) In Page 14, the 8th line from the bottom, the sentence “These molecules with unique shapes and configurations can also be used novel supramolecular synthons...” has revised as “These molecules with unique shapes and configurations can also be used as novel supramolecular synthons...” in the Revised Manuscript.

Reviewer 2:

The paper by Zhenyu Yang et al. describes the reaction of a hexaaldehyde substrate with two triamines to form various imine double-cages. The hexaaldehyde is conformationally semi-labile – it is partially rigidified by biaryl-type of connection of aromatic rings but it retains low-barrier rotations about single bonds. The unique feature of this hexaaldehyde is the possibility of forming two types of double clefts with the 3-fold symmetry that substantially differ by dimensions (among the unlimited number of other conformations). The authors reacted this hexaaldehyde with two aliphatic triamines and obtained imines by a reversible type of reaction. The short triamine formed only one type of a double-cage utilizing smaller clefts, while the longer triamine was able to produce two types of double-cages utilizing either small or larger clefts. For the obtained double-cages, interesting proton transporting properties have been reported, that rely on the high charge and porosity in the solid-state.

First of all, we would like to thank the Reviewer for carefully reviewing our manuscript and appreciating our research work.

Q1: Despite the fact that I think that the “dual-purpose” hexaaldehyde is interesting, and resulting double-cages may exhibit unique properties, I can’t agree with the “methodological” interpretation of the results that the authors provided. The authors gave a new name (“Conformation capturing”) to the well-known and intuitive phenomenon. It is quite typical that conformationally labile compounds, especially those with low activation energy barriers can easily adjust and form the products that thermodynamically most stable. It is also straightforward, that for a “small” partner, that does not fit to the large cleft only one type of product is formed while for the larger but still flexible partner two types of product are possible and formed. The idea of multidimensional reactivity that depends on the conditions (the reaction partner, template, or environment) and relies on size-match and conformational shape-adjustment constitutes the core of dynamic covalent chemistry. Therefore, the results presented in this paper are not “a novel yet simple size-matching strategy to identify and distinguish the interconverting conformational” a new strategy to “capture” conformers or to probe conformational space that “had remained unexplored” as the authors claim, but the typical and intuitive behavior of conformationally labile molecules participating in reversible reactions.

A1: Thanks for the reviewer’s comments. We agree with the reviewer’s opinion on “It is quite typical that conformationally labile compounds, especially those with low activation energy barriers can easily adjust and form the products that thermodynamically most stable”. However, as we calculated the conformation

energy of the two conformers of molecule **1** (Supplementary Fig. 9 and Table 1), we found that their energy difference is negligible (*ca.* 0.1 kJ mol⁻¹). It therefore indicates their fast interconversion, which makes it hard to say which conformation is thermodynamically more stable to pack into the corresponding crystal.

The reviewer stated that the conformation capturing is a well-known and intuitive phenomenon, and the comment itself is indeed correct but not in our case. First of all, we think the wording “large/small cleft” is somehow misleading, which has been revised as “As the formyl groups are permanently moving, the region with the probability of containing formyl moieties forms a circular ring within each cleft, and the range of the ring is different for **Conformer-1** and **-2** (middle column, Fig. 1).” (Page 3 in the Revised Manuscript, highlighted in yellow). Secondly, this is not intuitive as the reviewer thought to be. As shown in Fig. 1 in the Revised Manuscript, the average cleft size of **Conformer-1** is indeed larger (average radius $R = \frac{16.52-3.48}{2} + 3.48 = 10 \text{ \AA}$) than that of **Conformer-2** ($r = 9.52 \text{ \AA}$). If the intuitive phenomenon was true, the smaller partner (**CC-1**) would selectively capture the smaller cleft of **Conformer-2**. On the contrary, the molecular dynamics simulations and experiments both showed that larger **CC-2** fit both two conformers while smaller **CC-1** only captured **Conformer-1**. Therefore, the “size-matching strategy” in the manuscript is not a visual sense of cleft size in **Conformer-1** and **-2**, but the range of probability of formyl groups within the cleft, similar to that of electron cloud surrounding an atomic nucleus.

The rational of this design has been carefully explained in the Revised Supplementary Information, along with the revised Supplementary Fig. 10, give as follows:

As the formyl groups are permanently moving, the region with the probability of containing formyl moieties form a circular ring within each cleft, and the range of the cleft of **Conformer-1** and **-2** is different. The region with formyl motion is similar to that of electron cloud surrounding an atomic nucleus, which spans from 3.48 to 16.52 Å for **Conformer-1** and 6.76 to 12.28 Å for **Conformer-2**, respectively. We also calculated the sizes of **CC-1** and **-2**, which are 3.766 and 4.770 Å, respectively. Taken into consideration of the reactive distance between an amine and an aldehyde (esteemed to be 2.911 Å^{S22}), the effective sizes of **CC-1** and **-2** were calculated to be 6.677 and 7.681 Å, respectively. It therefore means that both **CC-1** and **-2** can easily cover the “formyl cloud” of **Conformer-1**, producing both *endo*-[1,2,4]diphane and *endo*-[1,2,5]diphane. On the other hand, only large-sized **CC-2** can theoretically reach all the formyl groups of **Conformer-2**, yielding the corresponding and *exo*-[1,2,5]diphane, while

these formyl moieties are out of reach for small-sized **CC-1**, unable to generate the corresponding *exo*-[1,2,4]diphane. This paragraph has been added in the Revised Supplementary Information.

Supplementary Figure 10. Model compound **1** exhibits infinite conformational isomers that fall into two types, namely **Conformer-1** with a pair of upper and lower clefts and **Conformer-2** with a pair of left and right clefts (right column). The region with the probability of containing formyl groups varies within the cleft of each conformer, determined by molecular dynamics simulations and illustrated with the projected circular rings that show the probability of formyl moieties (middle column). As the reactive distance of an amine and an aldehyde is esteemed to be 2.911 Å,^{S22} and the effective sizes of **CC-1** and **-2** were therefore calculated to be 6.677 and 7.681 Å, respectively (right column).

Q2: I think that the compounds that the authors synthesized are indeed interesting and create many new possibilities that can, in the future, become suitable for publication in *Nat. Commun.*, but, in this interpretation, the authors missed the point. Instead of focusing on the unique properties of the double cage products, which are way more interesting (but require more work and more examples), they described rather typical and intuitive behavior, presenting it as a new methodology. Therefore, I recommend the rejection of the paper in the current form. The results may become suitable for publication in the future, with the focal point changed and with more results about the obtained double cages.

A2: Thanks for the reviewer's comment. As we partially agree with reviewer's opinion the size-matching strategy seems quite straightforward, we think it is because our method is so straightforward that makes the Reviewer easily grasp the design and retrospectively considers the simplicity of the strategy (which was considered *is elegant yet simple* by **Reviewer 1** and *quite simple and relatively easy* by **Reviewer 3**). Following the suggestion of the Reviewer, we revised the focal point of the manuscript in the introduction by focusing on the simple method of preparation of a series of isomeric double cages with structural complexity, and we also emphasized the properties (proton conductivity) and structure-property correlation of the resulting double cages in the Revised Manuscript.

Indeed, we used the size-matching strategy to discover interesting structures related to the flexible conformations of a precursor, including but not limited to the three *diphanes* reported in the present work. For example, by employing the size-matching strategy, the precursor molecule **2** reacted with large-sized **CC-2** to selectively yield monofunctional **Cage-2** with a single cavity and one pendant -OH (Fig. C1a), which is already presented as a by-product in the original manuscript (Fig. 5 and Fig. C1a); on the other hand, the reaction between molecule **2** and small-sized **CC-1** selectively produced bifunctional **Cage-3** with two -OH at both ends of the molecule (Fig. C1a). We are currently polymerizing these two cages (as monomer) to form lateral and main-chain polycages, respectively, which will be reported in due course. Besides, we found that **Cage-2** (not **Cage-1**) self-assembled into superstructures with distinct four hierarchical levels (Fig. C2). The tertiary structure is supramolecular 2_1 -helix, and an array of these helices formed the quaternary superstructure.

The conformation tethering strategy also allowed us to prepare a series of chiral *diphanes* such as *R-endo*-[1,2,4]diphane, *S-endo*-[1,2,4]diphane, *R-endo*-[1,2,5]diphane and *S-endo*-[1,2,5]diphane (Fig. C1b). Taking *R-endo*-[1,2,4]diphane for instance, it also self-assembled into superstructures with four hierarchical levels (Fig. C3). It is worth noting that chiral channels were formed in the tertiary structures, which allows us to explore the induced chirality and spontaneous polarization of confined water. It endows the superstructure with interesting properties such as *non-linear optics* and *ferroelectrics*, which are under investigation in our lab.

Figure C1. Schematic representation of the synthesis procedure. (a) Selective synthesis of **Cage-2** and **Cage-3** with mono- and bifunctional groups via the size-matching strategy. Small-sized tris(2-aminoethyl)amine (**CC-1**) molecule produces **Cage-3** as the major product *via* dynamic reversible imination reaction and subsequent reduction, while **Cage-1** is the by-product (trace amount). By contrast, large-sized tris(3-aminopropyl)amine (**CC-2**) molecule yields **Cage-2** and **BFCage-4** as major and side product (trace amount), respectively. (b) The synthesis of ***R*-endo-[1,2,4]diphane** and ***S*-endo-[1,2,4]diphane**.

In a word, the strategy reported in the current manuscript provides a series of interesting molecular platform for the exploration of novel hierarchical superstructures, which allows us to search for emergent properties of these superstructures.

Figure C2. Hierarchical self-assembly of **Cage-2** during crystallization. a) Side view of its crystal structure (primary structure), b) supramolecular tetramer (secondary structure) self-assembled by four **Cage-2** molecules, c) supramolecular 2_1 -Helix self-organized by supramolecular tetramer, and d) the super lattice (quaternary structure) formed by an array of 2_1 -Helices. Hydrogen atoms and solvents are omitted for clarity, and the crystal belongs to orthorhombic space group $P2_1/n$.

Figure C3. Hierarchical self-assembly of *R-endo*-[1,2,4]diphane during crystallization. a) Side view of its crystal structure (primary structure), b) supramolecular dimer (secondary structure) self-assembled by two *R-endo*-[1,2,4]diphane molecules, c) supramolecular 3_2 -Helix self-organized by supramolecular dimer, forming a chiral intermolecular channel as shown in blue, and d) the super lattice (quaternary structure) formed by an array of 3_2 -Helices. Hydrogen atoms and solvents are omitted for clarity, and the crystal belongs to trigonal space group $P3_221$.

Q3: PXRD patterns – experimental and simulated are very different. Most likely more polymorphic forms are present in the samples.

A3: Thanks for the reviewer’s comment. This question has been addressed in the Q&A of the Reviewer 1. The previous results were indeed inconsistent, as the bulk samples used for PXRD were obtained directly from rotovap. In the Revised Manuscript, we carefully prepared the powder samples that were composed of large quantity of single crystals for *each diphane*, compared their PXRD patterns (both experimental and Pawley refinement) with the pattern simulated from SC-XRD data. As can be seen from Supplementary Fig. 13-15, the PXRD pattern of each sample fits well with the pattern simulated from SC-XRD data, with R_{wp} and R_p below 5%. These figures were newly collected, which have been amended and highlighted in yellow in the Revised Supplementary Information.

Supplementary Figure 13. Powder X-ray pattern of *endo*-[1,2,4]diphane, experimental pattern (*grey line*), pawley refinement (*blue circles*), simulated pattern (red line) and difference (*black line*) profiles for Pawley refinement ($R_{wp} = 3.56\%$, $R_p = 1.56\%$) at 298 K ($a = 13.717759$, $b = 15.453886$, $c = 16.448906$ Å, $P-1$). The reflection positions are marked with green.

Supplementary Figure 14. Powder X-ray pattern of *endo*-[1,2,5]diphane, experimental pattern (*grey line*), pawley refinement (*blue circles*), simulated pattern (red line) and difference (*black line*) profiles for Pawley refinement ($R_{wp} = 3.31\%$, $R_p = 3.93\%$) at 298 K ($a = 16.421116$, $b = 16.786190$, $c = 19.463487$ Å, $P-1$). The reflection positions are marked with green.

Supplementary Figure 15. Powder X-ray pattern of *exo*-[1,2,5]diphane, experimental pattern (*grey line*), pawley refinement (*blue circles*), simulated pattern (red line) and difference (*black line*) profiles for Pawley refinement ($R_{wp} = 3.36\%$, $R_p = 1.10\%$) at 298 K ($a = 16.303037$, $b = 16.787783$, $c = 19.356683$ Å, $P-1$). The reflection positions are marked with green.

Q4: It is quite typical that for the conformationally labile compound only one of the possible conformers is found in the crystal phase. So depriving the value of X-ray analysis by saying “SC-XRD, only able to identify Conformer-1 in solid-state” is not OK.

A4: Thanks for the reviewer’s comment, and the reviewer is indeed correct. In the original manuscript, we have already put a similar statement that “X-ray crystallography is a powerful method to unambiguously determine the absolute conformation of a molecule, but it often only distinguishes the conformer that provides the most efficient molecular packing.”, which agrees well with the reviewer’s opinion. In order to avoid the misleadingness, we revised the previous statement in Page 4 in the Original Manuscript to “Its single crystals suitable for X-ray crystallography were obtained by slow evaporation of solvent from its chloroform solution, and molecule **1** only exists the conformation corresponding to **Conformer-1** in the crystal structure (Fig. 2c), which is common for the conformationally labile compounds that often crystallize into the most efficient molecular packing.”, which has been highlighted in yellow in the Revised Manuscript (Line 10, Page 5).

Q5: The discussion part does not contain a discussion but a summary- so either some discussion should be added or the part should be called summary.

A5: Thanks for the reviewer’s comment. The title of the Discussion part in the Original Manuscript has been amended as Summary in the Revised Manuscript.

Q6: The authors should clearly state either the calculation/conclusions/discussion is for the imine or the amine in the protonated or non-protonated form because the torsional requirements and thus energies can be very different.

Answer: Thanks for the reviewer’s comment. First of all, we would like to differentiate the imine and amine form of *diphanes* with their definitions. As we described in the Revised Manuscript (the first two sentences of the last paragraph, Page 10): “The peaks in light blue represent the formation of the imine form of *endo*-[1,2,5]diphane, denoted *endo*-[1,2,5]diphane’, of which H_C is the proton of Schiff base and H_D the aromatic proton adjacent to H_C . The peaks in light red are assigned to the imine form of *exo*-[1,2,5]diphane, namely *exo*-[1,2,5]diphane’, of which $H_{C'}$ corresponds to the proton of Schiff base and $H_{D'}$ the aromatic proton in the immediate vicinity of $H_{C'}$.” We denoted the amine form of *diphanes* as *endo*-[1,2,4]diphane, *endo*-[1,2,5]diphane and *exo*-[1,2,5]diphane. Accordingly, their imine form are indicated as *endo*-[1,2,4]diphane’, *endo*-[1,2,5]diphane’ and *exo*-[1,2,5]diphane’.

In the calculation part of the manuscript (Fig. 1), the results were carried out by molecular dynamics simulations based on the formation of imine form of *diphanes*. In this section, we investigated the design principle of the size-matching strategy of *diphanes* construction, which should rely on reactive distance between formyl and amino groups for the effective formation of imine bonds. As we illustrated in the Supplementary Fig. 10, the effective size of CC-2 matches with the formyl activity range of Conformer-1 and -2 simultaneously, while CC-1 only matches Conformer-1. In the conclusion and discussion parts of the manuscript, all of the torsion related calculations were performed with amine form of *diphanes* without protonation, which were denoted as *endo*-[1,2,4]*diphane*, *endo*-[1,2,5]*diphane* and *exo*-[1,2,5]*diphane*.

Based on the reviewer's comment, we revised the sentence (Page 11 in the Revised Manuscript) "These results are in line with DFT calculations of SE and torsion angle of each *diphane*" as "These results are in line with DFT calculations of SE and torsion angle of each amine form of *diphane*", which was highlighted in yellow in the Revised Manuscript.

Q7: HPLC analysis of the distribution of products (after quenching) would be informative.

A7: Thanks for the reviewer's comment. Following the reviewer's suggestion, we performed the product distribution experiments with HPLC analysis. As seen in Supplementary Fig. 6, the crude reaction mixture of molecule 1 with small sized CC-1 only yielded on *diphane* species, which was confirmed by HPLC of the purified *endo*-[1,2,4]*diphane*; while the crude reaction mixture of molecule 1 with large sized CC-2 yielded both *endo*-[1,2,5]*diphane* and *exo*-[1,2,5]*diphane*, which were assigned with the purified products. This has been added and highlighted in yellow in the Revised Supplementary Information. The experimental details were added in the Revised Supplementary Information.

Supplementary Figure 6. Stacked HPLC traces of (a) the crude reaction mixture of molecule **1** with small sized CC-1, (b) *endo*-[1,2,4]diphane, (c) the crude reaction mixture of molecule **1** with large sized CC-2, (d) *exo*-[1,2,5]diphane (e) *endo*-[1,2,5]diphane.

Q8: The part starting from “No interchange between these two imine-based isomers has been observed throughout their formation process” and the conclusion that follows are not proved. The net result of the equilibrium does not change, but there are no proofs that the exchange processes are locked.

A8: Thanks for the reviewer’s comment. The reviewer is indeed correct, the NMR analysis only shows an apparent equilibrium that does not change. We therefore deleted the statement “No interchange between these two imine-based isomers has been observed throughout their formation process. It means that the erstwhile fast interconversion of conformers of molecule **1** was quenched upon reacting with CC-2, preventing the exchange of two isomers despite the dynamic feature of DCC”. (Page 10 in the Revised Manuscript)

Reviewer 3:

In this manuscript, Z. Yang et al. provides a novel strategy to differentiate and identify rapid interconverting conformational isomers which would be otherwise difficult to be detected by standard characterization techniques. The simple but interesting approximation consists of the idea to catch conformational isomers of a targeted system (precursor) by reacting them through dynamic covalent chemistry with molecular units (capturers) of different size. The strategy is demonstrated for a three-dimensional hexaformyl model precursor with an extremely-rich conformational space (*i.e.*, infinite conformers), which can be broadly classified into two distinct type of conformers. The outcome of the reported strategy (final species) not only allows the identification of two elusive conformers of the precursor but also provides new molecular cages with potential functionalities and applications (e.g., favorable proton conduction).

Answer: First of all, we would like to thank the Reviewer for carefully reviewing our manuscript and appreciating our research work. We appreciate the constructive suggestions and input from Reviewer 3, and the reviewer's point-by-point suggestions were used to tremendous improve our manuscript.

1. Within my limited knowledge about the literature of molecular cages, the proposed synthetic method (key result of the study) looks quite simple and relatively easy to be used for the design of novel 3D molecular cages. It is difficult, however, to me to assess the significance and broad scope of the proposed approach for a general scientific community in the frontier of chemistry, materials chemistry and chemical biology. Nevertheless, the relevance of the synthetic procedure for the design of novel and exotic molecular cages can be strengthened and highlighted with a more extended introduction (only a few sentences) emphasizing the advantages of this new approach with respect to the previous ones to obtain organic molecular cages.

Answer: Thanks for the reviewer's comment. The novelty of this paper is not only about a new approach of synthesizing cages, but also about the concept of conformer capturing in solution through chemical bonds formation *via* size-matching strategy, which allowed easy structural inspection by conventional characterization, and also led to the discovery of novel structures that were previously inaccessible. Regarding the advantages of our synthetic procedure compared to the previous ones for the design of novel cages, we successfully constructed two exotic organic cages with sandglass-shaped geometry in a simple way and high yield. Before our work, only one similar pure organic cage structure with dumbbell geometry and twin cavities was reported by Greenaway and Cooper in *Chem.*

Eur. J. **26**, 3718–3722 (2020), which had low synthetic efficiency. As we stated on Page 2 of the Revised Manuscript “A variety of cages with different three-dimensional shapes so far have been elaborated, most of which exhibit a single cavity, with the exception of dumbbell-shaped cages with twin cavities recently reported by separate works of Li and Stang¹³ and Cooper.¹⁴”. To obtain the organic cages with twin cavities in a more efficient way, we rationally designed the three-dimensional hexaformyl precursor **1**, which theoretically exhibits infinite conformers in solution. Aided by molecular dynamics simulations, we selected two triamino cap-shaped molecules with different sizes to capture different conformers of the precursor **1** in solution *via* a size-matching strategy. In a word, using this method, we not only captured the different conformers of the precursor **1** in solution, which are challenging to identify experimentally with traditional techniques, but also obtained two novel sandglass-shaped organic cages in a highly efficient way.

2. The study is very well-organized, well-written and can be easily followed with the main idea clear along the manuscript. I have really enjoyed the paper. Nonetheless, a criticism, related with the previous one (significance), would be to provide a sufficient context with more comparisons with previous works to make clear the relevance of the study.

Answer: Thanks for the reviewer’s comment. We have revised introduction section in the Revised Manuscript by emphasizing the significance of the manuscript in the area of covalent organic cages and their application as self-assembling building blocks for the search of novel supramolecular materials with emergent functions. It is given as follows: “Covalent organic cages with well-defined intrinsic porosity have attracted increasing attention for the last decade.¹⁻⁸ Their internal cavity and external channels have been applied for selective recognition/separation,^{2-3,9} catalysis,¹⁰⁻¹¹ sensing,¹² and so on. A variety of cages with different three-dimensional shapes so far have been elaborated, most of which exhibit a single cavity, with the exception of dumbbell-shaped cages with twin cavities recently reported by separate works of Li and Stang¹³ and Cooper.¹⁴ Besides, cages with different geometries could be employed as powerful building blocks in search of novel supramolecular materials with emergent properties that are otherwise inaccessible by conventional molecules;¹⁵ however, this has been largely overlooked”.

3. Concerning validity of the study, the experimental findings seem to be solid and support the conclusions drawn. Nevertheless, I believe that the theoretical part, which is quite relevant in the study, can be improved. In this respect, I would suggest the following points to be considered:

- 1) The experimental analysis (H NMR) about the conformational interconversion of 1 is performed in solution. However, DFT calculations seem to be performed in gas phase. That effect may have an effect on the energy difference between **Conformer-1** and **-2**. Therefore, I would include solvent effects in the calculations (a continuum model can be enough for this study).

Answer: Thanks for the reviewer's comment. We recalculated the free energy of each isomer (**conformer-1** and **conformer-2**) with geometry optimization using DFT calculation (B3LYP-D3 in PCM with chloroform as solvent). The PCM solvation model was used to deal with the solvation effect. The results showed that the energy difference of the two conformers in chloroform is even lower (*ca.* 0.1 kJ mol⁻¹) compared to the corresponding gas phase, indicating their fast interconversion.

Supplementary Table 1. The calculated conformation energy with B3LYP and M062X functionals.

	B3LYP/6-31G* (Gas)	B3LYP-D3/6-31G* (CHCl ₃)	B3LYP-D3/6-31G* (CHCl ₃ , Free energy)	M062X-D3/6-31G* (CHCl ₃)	M062X-D3/6-31G* (CHCl ₃ , Free energy)
Conformer-1	0.00	0.00	0.00	0.00	0.00
Conformer-2	-1.13	0.40	-0.12	0.45	0.72

- 2) van der Waals (vdW) intramolecular interactions can also have an effect on the energy difference between **conformer-1** and **-2**. Plain B3LYP functional cannot capture vdW interactions and should be augmented with external approximations (the most practical would be the Grimme's D3 approach). I would re-do the calculations at least at the B3LYP-D3/6-31G* level to check that intramolecular vdW interactions are not playing an important effect.

Answer: Thanks for the reviewer's comment. Following the reviewer's advice, we also have redone the calculations at the B3LYP-D3/6-31G* level. The results showed that the energy difference of the two conformers is 0.4 kJ mol⁻¹, indicating the intramolecular vdW interactions do not play an important effect on the energy difference between **conformer-1** and **-2**.

Supplementary Table 1. The calculated conformation energy with B3LYP and M062X functionals.

	B3LYP/6-31G* (Gas)	B3LYP-D3/6-31G* (CHCl ₃)	B3LYP-D3/6-31G* (CHCl ₃ , Free energy)	M062X-D3/6-31G* (CHCl ₃)	M062X-D3/6-31G* (CHCl ₃ , Free energy)
Conformer-1	0.00	0.00	0.00	0.00	0.00
Conformer-2	-1.13	0.40	-0.12	0.45	0.72

- 3) As the energy difference between **conformer-1** and **-2** is quite small even in the range of error of the DFT method, I would check that the energy order is preserved with another density functional. In this sense, extra calculations with the M06-2X, which can describe vdW interactions at short range, can be a good alternative to check that the theoretical results are robust.

Answer: Thanks for the reviewer’s comment. We recalculated the free energy of each isomer (**conformer-1** and **conformer-2**) with another M06-2X-D3 hybrid functional to check whether the energy order is preserved. As shown in the Supplementary Table 1 in the Revised Supplementary Information, although the energy order is different for B3LYP-D3 and M062X-D3 functionals, the overall deviation between them is quite small. Thus, the results reported in this paper are all calculated by B3LYP-D3 method in chloroform unless the specific instructions.

Supplementary Table 1. The calculated conformation energy with B3LYP and M062X functionals.

	B3LYP/6-31G* (Gas)	B3LYP-D3/6-31G* (CHCl ₃)	B3LYP-D3/6-31G* (CHCl ₃ , Free energy)	M062X-D3/6-31G* (CHCl ₃)	M062X-D3/6-31G* (CHCl ₃ , Free energy)
Conformer-1	0.00	0.00	0.00	0.00	0.00
Conformer-2	-1.13	0.40	-0.12	0.45	0.72

- 4) In the caption of Fig. 2a, the authors write: “Fast interconversion of **Conformer-1** and **-2** with extremely low isomerization barrier.” However, the authors have not computed the isomerization barrier, they provide an energy difference between both conformers. An estimate of the low isomerization barriers can provide by calculating the torsional potential through: i) the rotation of the carbon-carbon bond of the central benzene ring and the subsequent sp³ carbon atom and ii) the rotation around the inter-ring carbon carbon bond of the two aryl rings within a peripheral arm. Then the author say: “Free energy of each isomer was determined by geometry optimization

(DFT, B3LYP/6-31G*)”. I was wondering if the energy provided is actually a free energy difference computed from vibrational calculations or is simply the energy difference calculated between the optimized **conformer 1** and **2**. If it is obtained by the latter approach is not a free energy estimate.

Answer: Thanks for the reviewer’s comments.

(1) We probed the isomerization barrier by calculating the torsional potential shown in Supplementary Fig. 9 in the Revised Supplementary Information: i) The rotation of the carbon-carbon bond of the central benzene ring and the subsequent sp^3 carbon atom (showed in black line). ii) The rotation around the inter-ring C-C bond of the two aryl rings within a peripheral arm (showed in red line), and the results showed that the rotation of bonds **2** needed to overcome much lower torsion potentials than the corresponding bonds **1**, which clearly verified our statement “fast interconversion of **Conformer-1** and **-2** with extremely low isomerization barrier.”

Supplementary Figure 9. Variation of conformation energy with 1) the rotation of the carbon-carbon bond of the central benzene ring and the subsequent sp^3 carbon atom and 2) the rotation around the inter-ring carbon-carbon bond of the two aryl rings within a peripheral arm.

(2) The energy of each conformers provided in Fig. 2a is free energy, which was recalculated by geometry optimization using DFT calculation (B3LYP-D3 in PCM with chloroform as solvent). In addition, we revised the caption of Fig. 2 in the Revised Manuscript “(a) Fast interconversion of **Conformer-1** and **-2** with extremely low isomerization barrier. Free energy of each isomer was determined by geometry optimization (DFT, B3LYP/6-31G*)” as “(a) Fast interconversion of **Conformer-1** and **-2** with extremely low isomerization barrier. Free energy of each isomer was determined by geometry optimization using DFT calculation (B3LYP-D3 in PCM with

chloroform as solvent)”).

- 5) In the Supplementary Information (DFT calculations section), the authors define the strain energy as the energy difference between the EC unit found in the cage and that obtained after optimization. The abbreviation EC is not defined.

Answer: Thanks for the reviewer’s comment. Sorry for our negligence, in the Revised Supplementary Information (DFT calculations section), the sentence “It was defined as the energy difference between the EC unit in the cage and the optimized EC monomers. Here, it should be noted the end group of the EC unit in the cage was reduced to aldehyde structure to maintain the equality of the calculations.” have been revised as “It was defined as the energy difference between the conformers in the cage and the optimized free conformer. It should be noted the end group of the conformer in the cage was reduced to aldehyde structures to maintain the equality of the calculations.” We therefore do not use EC anymore in the Revised Supplementary Information for the sake of clarity.

- 6) In the Supplementary Information, there is a section devoted to Molecular dynamics simulations. However, the authors do not call to this section during the discussion in the main text which is weird. On the other hand, I do not understand the histograms in Supplementary Figure 4, is it possible that they are interchanged? If it is the case, it would have more sense to me. Nevertheless, I think that Supplementary Figure 4 deserves a brief discussion at least in the Supplementary Information.

Answer: Thanks for the reviewer’s comments. Actually, Fig. 1 presents a concept summary based on Supplementary Fig. 4 in the original manuscript, which is now Supplementary Fig. 10 in the Revised Supplementary Information.

In Page 4 in the Revised Manuscript we added “As briefly mentioned above, the geometry optimizations determined by DFT calculation (B3LYP-D3 in PCM with chloroform as solvent, see Supplementary Information for detailed calculation)” and “The energy difference of the two conformers is extremely low (*ca.* 0.1 kJ mol⁻¹, Supplementary Table 1 and Supplementary Fig. 9)” to call the molecular dynamic simulations. As suggested by the Reviewer, we also discussed the rationale of this paper with Supplementary Fig. 10 in the Revised Supplementary Information. Please refer to it for detailed discussion.

- 7) Apart from the strain energies evaluated for the four diphane cages, the relative difference between the two types of cages (*exo* and *endo*) should be provided to see if they can correlate with the experimental yield obtained for the final cages.

Answer: Thanks for the reviewer's comments. The Reviewer is indeed correct. For example, as we already put it in the original manuscript, "We also noticed that the conversion rates of *endo*- and *exo*-[1,2,5]diphane' were different, as the former was generated faster than the latter, leading to a product distribution of 75% *endo*- and 25% *exo*-[1,2,5]diphane', respectively". This is in line with the strain energy of the *diphanes*.

- 8) Typo in the caption of Fig. 4 (second line) "truncatcd" should be substituted by "truncated".

Answer: Thanks for the reviewer's comments. In Page 8 of the Revised Manuscript, the word "trancated" in the caption of Fig. 4 (second line) has been revised as "truncated".

- 9) As a future suggestion (not for this paper), it would be great if you are able to have control and break the final molecular cage by external stimuli but isolating the conformational isomers.

Answer: Thanks for the reviewer's comments. Capture conformers in solution in a controllable and reversible way is our ultimate goal and we are making efforts to achieve it in the near future.

Changes List for Manuscript NCOMMS-21-02237 and Supplementary Information

➤ In manuscript

1. In Page 1 of the Original Manuscript, the second line of the abstract, the word “fileds” has been revised as “fields” in the Revised Manuscript (Page 1, the second line of the abstract).
2. In Page 1 of the Original Manuscript, line 7 of the abstract, “DFT calculations” has been revised as “molecular dynamics simulations” in the Revised Manuscript (Page 1, line 7 of the abstract).
3. In Page 1 of the Original Manuscript, line 3 from the bottom of the abstract, “ $2.91 \times 10^{-4} \text{ S cm}^{-1}$ ” has been revised as “ $1.37 \times 10^{-5} \text{ S cm}^{-1}$ ” in the Revised Manuscript (Page 1, line 3 from bottom of the abstract).
4. In Page 1 of the Original Manuscript, line 2 from the bottom of the abstract, “ 3×10^4 ” has been revised as “ 10^3 ” in the Revised Manuscript (Page 1, line 2 from the bottom of the abstract).
5. In Page 1 of the Original Manuscript, the bottom line of the abstract, “ 10^3 ” has been revised as “ 10^4 ” in the Revised Manuscript (Page 1, the bottom line of the abstract).
6. In Page 2 of the Original Manuscript, the first two paragraphs of the introduction have been deleted and revised as “Covalent organic cages with well-defined intrinsic porosity have attracted increasing attention for the last decade.¹⁻⁸ Their internal cavity and external channels have been applied for selective recognition/separation,^{2-3,9} catalysis,¹⁰⁻¹¹ sensing,¹² and so on. A variety of cages with different three-dimensional shapes so far have been elaborated, most of which exhibit a single cavity, with the exception of dumbbell-shaped cages with twin cavities recently reported by separate works of Li and Stang¹³ and Cooper.¹⁴ Besides, cages with different geometries could be employed as powerful building blocks in search of novel supramolecular materials with emergent properties that are otherwise inaccessible by conventional molecules;¹⁵ however, this has been largely overlooked.” in the Revised Manuscript (Page 2, last paragraph).
7. In Page 3 of the Original Manuscript, the fourth line of the caption in Fig. 1, “DFT calculation (B3LYP/6-31G*)” has been revised as “molecular dynamics simulations” in the Revised Manuscript (Page 4, the fourth line of the caption in Fig. 1).
8. In Page 3 of the Original Manuscript, line 5 of the caption in Fig. 1, the sentence “which is used to quantify the size of the clefts” has been revised as “which is used to quantify the range covered by

permanently moving formyl groups in each cleft” in the Revised Manuscript (Page 4, line 5 of the caption in Fig.1).

9. In Page 3 of the Original Manuscript, line 8 of the caption in Fig.1, “DFT” has been revised as “molecular dynamics simulations” in the Revised Manuscript (Page 4, line 9 of the caption in Fig.1).
10. In Page 3 of the Original Manuscript, line 4 from the bottom, “density-functional theory (DFT) calculation” has been revised as “molecular dynamics simulations” in the Revised Manuscript (Page 3, line 5).
11. In Page 3 of the Original Manuscript, line 3 from the bottom, the sentence “This hexaformyl molecule **1** exhibits all possible conformations that fall into two distinct types” has been revised as “This conformationally labile molecule **1** with six formyl groups exhibits all instantly interchanging conformations that fall into two distinct types” in the Revised Manuscript (Page 3, line 6).
12. In Page 4 of the Original Manuscript, the sentences of the first paragraph “Due to the molecular motion, the size of these clefts varies spontaneously, with the permanently moving formyl groups forming a circular ring (middle column, Fig. 1). As the size range of the clefts of **Conformer-1** and **-2** is different...” have been revised as “As the formyl groups are permanently moving, the region with the probability of containing formyl moieties forms a circular ring within each cleft, and the range of the ring is different for **Conformer-1** and **-2** (middle column, Fig. 1). By virtue of their low isomerization barrier, their identification poses a formidable challenge for conventional methods with low time-scale resolution, such as nuclear magnetic resonance (NMR) and infrared (IR) spectroscopies. X-ray crystallography is a powerful method to unambiguously determine the absolute conformation of a molecule, but it often only distinguishes the conformer that provides the most efficient molecular packing.²²” in the Revised Manuscript (Page 3, line 9).
13. In Page 4 of the Original Manuscript, at the end of the first paragraph, we added the sentences of “Moreover, these three *diphanes* self-assembled into similar superstructures; depending on the width of the intramolecular channels, they exhibit dramatic different capacity of proton conduction, which is distinct by an order of magnitude of 10^4 .” in the Revised Manuscript (Page 3, at the end of the last paragraph).
14. In Page 4 of the Original Manuscript, the first sentence of the second paragraph “As briefly mentioned above, the geometry optimizations (DFT, B3LYP/6-31G*) showed molecule **1** exists infinite forms

of conformers *via* all possible C-C single bond rotations, from which two types of conformers, **Conformer-1** and **-2**, were identified (Fig. 1 and Fig. 2a).” has been revised as “As briefly mentioned above, the geometry optimizations determined by DFT calculation (B3LYP-D3 in PCM with chloroform as solvent, see Supplementary Information for detailed calculation) showed that molecule **1** exists infinite forms of conformers *via* all possible C-C single bond rotations, from which two types of conformers, **Conformer-1** and **-2**, were identified (Fig. 1 and Fig. 2a, Supplementary Fig. 9).” in the Revised Manuscript (Page 4, line 6 from the bottom).

15. In Page 4 of the Original Manuscript, line 8 from the bottom “(ca. 1.1 kJ mol⁻¹)” has been revised as “(ca. 0.1 kJ mol⁻¹, Supplementary Table 1 and Supplementary Fig. 9)” in the Revised Manuscript (Page 5, line 2).
16. In Page 4 of the Original Manuscript, line 7 from the bottom, the sentence “such as variable-temperature NMR²²⁻²³ and time-resolved spectroscopies.²⁴⁻²⁵” was added in the Revised Manuscript (Page 5, line 4).
17. In Page 4 of the Original Manuscript, line 3 from the bottom, the word “spectroscopies” has been revised as “spectroscopy” in the Revised Manuscript (Page 5, line 9).
18. In Page 4 of the Original Manuscript, the last sentence “Its single crystals suitable for X-ray crystallography were obtained by slow evaporation of solvent from its chloroform solution, but only **Conformer-1** could be identified (Fig. 2c).” has been revised as “Its single crystals suitable for X-ray crystallography were obtained by slow evaporation of solvent from its chloroform solution, and molecule **1** only exists the conformation corresponding to **Conformer-1** in the crystal structure (Fig. 2c), which is common for the conformationally labile compounds that often crystallize into the most efficient molecular packing.²⁶” in the Revised Manuscript (Page 5, the last sentence of the second paragraph).
19. In Page 5 of the Original Manuscript, the third line in the caption of Fig. 2 “Free energy of each isomer was determined by geometry optimization (DFT, B3LYP/6-31G*)” has been revised as “Free energy of each isomer was determined by DFT calculation (B3LYP-D3 in PCM with chloroform as solvent)” in the Revised Manuscript (Page 5, the third line in the caption of Fig. 2).
20. In Page 5 of the Original Manuscript, the first sentence “Covalent organic cages have attracted increasing interest in the recent decade,¹⁴ and their porosity and geometric diversity have widely been

explored for selective recognition/separation,¹⁷ catalysis,¹⁸ hierarchical self-assembly,¹⁹ etc.” has been deleted as shown in the Revised Manuscript (Page 6, the first paragraph).

21. In Page 6 of the Original Manuscript, line 6 from the bottom “The single crystals suitable for X-ray analysis were obtained by slow evaporation of its THF solution with additional trifluoroacetic acid (TFA) for better solubility. It crystallized into...” has been revised as “The single crystals suitable for X-ray analysis were obtained by slow evaporation of **endo-[1,2,4]diphane** solution in THF with additional trifluoroacetic acid (TFA) for better solubility. **endo-[1,2,4]Diphane** crystallized into...” in the Revised Manuscript (Page 7, the first line).
22. In Page 7 of the Original Manuscript, the word “are” in the third line has been revised as “is” in the Revised Manuscript (Page 7, the third line from the bottom).
23. In Page 7 of the Original Manuscript, in line 5 we added the sentence “In addition, the X-ray structure of **endo-[1,2,4]diphane** showed that the tertiary amines on the aliphatic chains were not protonated, even with the presence of TFA. This is presumably due to the steric hindrance of the irregularly stretched amine moiety caused by the rigid strain conformation, as well as the electrostatic repulsion of the neighbouring TFA molecules (Supplementary Fig. 12).” in the Revised Manuscript (Page 7, the second line from the bottom).
24. In Page 7 of the Original Manuscript, the second line of the caption in Fig. 4, the word “trancated” has been revised as “truncated” in the Revised Manuscript (Page 8, the second line of the caption in Fig. 4).
25. In Page 8 of the Original Manuscript, the second line, “Supplementary Fig. 4” has been revised as “Supplementary Fig. 10” in the Revised Manuscript (Page 8, the last sentence)
26. In Page 8 of the Original Manuscript, line 12, the sentence “This dumbbell-shaped cage had been unprecedented until the very recent and separate works of Li and Stang,^{16a} and Greenaway and Cooper.^{16b}” has been deleted in the Revised Manuscript (Page 9, line 10).
27. In Page 8 of the Original Manuscript, line 3 of the third paragraph, “as above” has been deleted in the Revised Manuscript (Page 9, line 12 from the bottom).
28. In Page 8 of the Original Manuscript, line 4 from the bottom, “52.26 vs. 41.56” has been revised as “48.00 vs. 36.96” in the Revised Manuscript (Page 9, line 4 from the bottom).

29. In Page 8 of the Original Manuscript, line 3 from the bottom, “172.5°” has been revised as “174.66°” in the Revised Manuscript (Page 9, line 3 from the bottom).
30. In Page 8 of the Original Manuscript, line 2 from the bottom, “177.2°” has been revised as “176.57°” in the Revised Manuscript (Page 9, line 2 from the bottom).
31. In Page 9 of the Original Manuscript, line 2 of the caption in Fig. 5 “(DFT, B3LYP/6-31G*)” has been revised as “(DFT, B3LYP-D3 in PCM with chloroform as solvent)” in the Revised Manuscript (Page 10, line 2 of the caption in Fig. 5).
32. In Page 9 of the Original Manuscript, line 5 from the bottom, the sentence “No interchange between these two imine-based isomers has been observed throughout their formation process. It means that the erstwhile fast interconversion of conformers of molecule **1** was quenched upon reacting with **CC-2**, preventing the exchange of two isomers despite the dynamic feature of DCC.” has been deleted in the Revised Manuscript (Page 10, line 5 of the bottom).
33. In Page 10 of the Original Manuscript, the third line, “Supplementary Fig. 9” has been revised as “Supplementary Fig. 8” in the Revised Manuscript (Page 11, the third line).
34. In Page 10 of the Original Manuscript, the last paragraph “These results are in line with DFT calculations of SE and torsion angle of each *diphane* (Supplementary Fig. 5). The SE of *exo*-[1,2,4]diphane is the highest with 99.85 kJ mol⁻¹, larger than 92.32 kJ mol⁻¹ for *endo*-[1,2,4]diphane, and *exo*-[1,2,4]diphane is significantly bent with the smallest torsion angle of $\tau_4 = 172.5^\circ$, smaller than 174.0° for *endo*-[1,2,4]diphane. When using the large-sized **CC-2**, the SE values of the resulting *diphanes* are considerably lower, with 65.86 kJ mol⁻¹ for *endo*-[1,2,5]diphane, and 77.98 kJ mol⁻¹ for *exo*-[1,2,5]diphane, respectively, echoed with their corresponding torsion angles.” has been revised as “These results are in line with DFT calculations of SE and torsion angle of each amine form of *diphane* (Supplementary Fig. 11). The SE of *exo*-[1,2,4]diphane is the highest with 102.13 kJ mol⁻¹, larger than 96.38 kJ mol⁻¹ for *endo*-[1,2,4]diphane, and *exo*-[1,2,4]diphane is significantly bent with the smallest torsion angle of $\tau_4 = 174.12^\circ$, smaller than 175.04° for *endo*-[1,2,4]diphane. When using the large-sized **CC-2**, the SE values of the resulting *diphanes* are considerably lower, with 67.33 kJ mol⁻¹ for *endo*-[1,2,5]diphane, and 77.00 kJ mol⁻¹ for *exo*-[1,2,5]diphane, respectively, echoed with their corresponding torsion angles.” in the Revised Manuscript (Page 11, the last paragraph).

35. In Page 11 of the Original Manuscript, the table of Fig. 7 has been revised in the Revised Manuscript (Page 12, the table of Fig. 7).
36. In Page 11 of the Original Manuscript, the first line of the caption in Fig. 7, the sentence “The comparison of molecular structure of three *diphanes*” has been revised as “The comparison of molecular structure of three *diphanes* with the presence of TFA” in the Revised Manuscript (Page 12, the first line of the caption in Fig. 7).
37. In Page 11 of the Original Manuscript, line 8 of the caption in Fig. 7, “at 303 and 343 K” has been revised as “at 308 and 338 K” in the Revised Manuscript (Page 12, line 9 of the caption in Fig. 7).
38. In Page 12 of the Original Manuscript, the first paragraph, the sentences “We therefore encapsulated trifluoroacetic acids (TFA) into the crystalline superstructures of the three *diphanes*, and evaluated their performances of proton conduction (Fig. 7). As determined by SC-XRD, **endo-[1,2,4]diphane** and **endo-[1,2,5]diphane** exhibit similar geometry, *i.e.* sandglass-shaped structure (Fig. 7a and c). However, their self-assembled superstructures are remarkably different. **endo-[1,2,4]Diphane** molecules are closely packed, and it provides no clear transportation channels for TFA molecules (Fig. 7b), which are trapped within the intrinsic and extrinsic cavities of the *diphanes*. In contrast, **endo-[1,2,5]diphanes** self-assemble into a crystalline phase with ordered channels (highlighted with red column) that allow the mobility of TFA molecules (Fig. 7d). **exo-[1,2,5]Diphane** adopts a dumbbell-shaped geometry, and its corresponding superstructure also exhibits TFA channels (Fig. 7f). Compared to the superstructure of **endo-[1,2,5]diphane** with ordered channel ($d_1 = 6.7 \text{ \AA}$), **exo-[1,2,5]diphane** molecules are packed more tightly, and form zig-zag and narrower channels ($d_2 = 4.7 \text{ \AA}$), which might hinder the mobility of TFA molecules.” has been revised as “We therefore encapsulated trifluoroacetic acids (TFA) into the crystalline superstructures of the three *diphanes* (denoted as *diphanes-TFA*), and evaluated their performances of proton conduction (Fig. 7). As determined by SC-XRD, **endo-[1,2,4]diphane** and **endo-[1,2,5]diphane** exhibit similar geometry, *i.e.* sandglass-shaped structure (Fig. 7a and c). However, their self-assembled superstructures are remarkably different. **endo-[1,2,4]Diphane** molecules are closely packed, and it provides no clear transportation channels for protons hopping (Fig. 7b), where TFA molecules are trapped within the intrinsic and extrinsic cavities of the *diphanes*. In contrast, **endo-[1,2,5]diphanes** self-assemble into a crystalline phase with ordered channels (highlighted with red column) that allow the transportation

of protons (Fig. 7d). *exo*-[1,2,5]Diphane adopts a dumbbell-shaped geometry, and its corresponding superstructure also exhibits protons channels (Fig. 7f). Compared to the superstructure of *endo*-[1,2,5]diphane with ordered channel ($d_1 = 6.7 \text{ \AA}$), *exo*-[1,2,5]diphane molecules are packed more tightly, and form zig-zag and narrower channels ($d_2 = 4.7 \text{ \AA}$), which might hinder the transportation of protons.” in the Revised Manuscript (Page 13, from the third line).

39. In Page 12 of the Original Manuscript, the second paragraph has been revised in the Revised Manuscript (Page 13, the last paragraph).
40. In Page 12 of the Original Manuscript, the title “Discussion” has been revised as “Summary” in the Revised Manuscript (Page 14).
41. In Page 12 of the Original Manuscript, the last line “DFT calculation” has been revised as “molecular dynamics simulations” in the Revised Manuscript (Page 14, line 5 of the Summary section).
42. In Page 13 of the Original Manuscript, the line 4 of the second paragraph, “as” has been added in the Revised Manuscript (Page 14, line 7 from the bottom).
43. In Page 15 of the Original Manuscript, the “Single-crystal X-ray diffraction” part has been revised in the Revised Manuscript (Page 16).
44. In Page 15 of the Original Manuscript, the “Powder X-ray diffraction” part has been revised in the Revised Manuscript (Page 17).
45. In Page 16 of the Original Manuscript, in the “Acknowledgments” section, “Bona Dai (ssNMR)” has been added in the Revised Manuscript (Page 17).
46. In Page 16 of the Original Manuscript, in the “Author contributions” section, “Z.Y. and S.Z. conceived and designed this work.” has been revised as “Z.Y. and C.Y. contributed equally to this work.” in the Revised Manuscript (Page 17).
47. In Page 16 of the Original Manuscript, the “References” part has been totally revised in the Revised Manuscript (Page 18).

➤ **In Supplementary Information**

1. In Page S2 of the Original Supplementary Information, the table of contents and the order of titles have been revised in the Revised Supplementary Information (Page S2).
2. In Page S4 of the Original Supplementary Information, “**Solid-state nuclear magnetic resonance (ssNMR)** Solid-state ^1H MAS single-pulse and DQ/SQ spectra were recorded on a Bruker AVANCE NEO 600 MHz, 3.2 mm rotor, MAS of 15 MHz, recycle delay of 5 sec.” has been added in the Revised Supplementary Information (Page S4, the first line).
3. In Page S4 of the Original Supplementary Information, the content of “**Single crystal X-ray diffraction (SC-XRD)**” has been revised in the Revised Supplementary Information (Page S4, the last paragraph).
4. In Page S4 of the Original Supplementary Information, the content of “**Powder X-ray diffraction (PXRD)**” has been revised in the Revised Supplementary Information (Page S5, the first paragraph).
5. In Page S4 of the Original Supplementary Information, “**High performance liquid chromatography (HPLC)** HPCL analysis were performed on a Shimadzu LC-20AD instrument at room temperature using a Daicel Chiralcel IA column. The elution was employed within 60 min with 18% CHCl_3 in ethanol containing 0.5% diethylamine of total volume at a flow rate of 0.6 mL/min. The sample concentration was 5.0 mM in methanol, and the injection volume was 10 μL . The absorbance of *endo*-[1,2,4]diphane was monitored at 254 nm, *endo*-[1,2,5]diphane and *exo*-[1,2,5]diphane were monitored at 298 nm.” has been added in the Revised Supplementary Information (Page S5, the last paragraph).
6. In Page S7 of the Original Supplementary Information, line 3 from the bottom, “HR-MS (MALDI): $\text{C}_{73}\text{H}_{49}\text{O}_5$ ($\text{M}+\text{H}$) $^{+}$ ” of molecule **1** has been revised as “HR-MS (MALDI): $\text{C}_{73}\text{H}_{49}\text{O}_5$ (M) $^{+}$ ” in the Revised Supplementary Information (Page S8, line 3 from the bottom).
7. In Page S9 of the Original Supplementary Information, the third line, the yield of the product **2b** (2.0 g, 86%) has been revised as **2b** (2.2 g, 96%) in the Revised Supplementary Information (Page S10, the third line).
8. In Page S9 of the Original Supplementary Information, the yield of compound **2** (1.8 g, 85%) has been revised as compound **2** (1.4 g, 67%), and its “HR-MS (MALDI): (MALDI): $\text{C}_{33}\text{H}_{23}\text{O}_3^+$ ($\text{M}+\text{H}$) $^{+}$ ”

has been revised as “HR-MS (MALDI): (MALDI): $C_{33}H_{23}O_3^+ (M)^+$ ” in the Revised Supplementary Information (Page S10).

9. In Page S11 of the Original Supplementary Information, the “**Theoretical calculations**” section has been revised in the Revised Supplementary Information (Page S18).
10. In Page S13 of the Original Supplementary Information, the “**Strain Energy (SE) and Torsion Angles of *diphanes* calculations**” section has been revised in the Revised Supplementary Information (Page S21).
11. In Page S16 of the Original Supplementary Information, the “**HPLC analysis of the distribution of the *diphanes***” section has been added in the Revised Supplementary Information (Page S15).
12. In Page S17 of the Original Supplementary Information, the caption in Supplementary Figure 8 has been revised in the Revised Supplementary Information (Page S16).
13. In Page S19 of the Original Supplementary Information, the “**Proton conductivity**” part has been revised in the Revised Supplementary Information (Page S30).
14. In Page S21 of the Original Supplementary Information, the “**Powder X-ray diffraction pattern**” part has been revised in the Revised Supplementary Information (Page S29).
15. In Page S22 of the Original Supplementary Information, the “**X-ray crystallography**” part has been revised in the Revised Supplementary Information (Page S23).
16. In Page S26 of the Original Supplementary Information, the **Supplementary Figure. 12** has been added in the Revised Supplementary Information (Page S28).
17. In Page S26 of the Original Supplementary Information, “**Solid-state NMR spectroscopy (ssNMR)**” section has been added in the Revised Supplementary Information (Page S34).
18. In Page S46 of the Original Supplementary Information, the “**References**” has been revised in the Revised Supplementary Information (Page S56).

REVIEWER COMMENTS

Reviewer #1 (Remarks to the Author):

I am pleased to have read the revised version of the manuscript by Professor Zhang and co-workers which has gone through a substantial edit and as a result has been significantly improved. The PXRD part of the new version of the manuscript has been thoroughly revised and I am glad to see that a correlation between the simulated and recorded powder data now exists. The peak widths of the experimental patterns are rather large considering that single crystals were used in data collection. This could be explained, however, by partial desolvation of the crystals if the capillaries used in data collection were loaded with dried crystals and not together with mother liquor. This small detail, i.e. were the capillaries loaded with or without crystallization solvent, is an important information and should be added in the experimental detail in the supporting information to avoid confusion.

In addition, the following minor errors have been introduced into the SI material during the revision process, which should be addressed:

- 1) The given densities in single crystal X-ray diffraction data tables (supplementary tables 2-6) should be in g/cm^3 (not in M g/cm^3).
- 2) Check the graph legend in Supplementary Figure 14, which has the descriptions of the different graphs in wrong order.

In overall, the X-ray structure determination and crystallographic analyses are expertly done and after the above minor corrections have been made, I am pleased to support the publication of the manuscript in terms of the quality of its crystallographic part.

Reviewer #2 (Remarks to the Author):

Despite numerous arguments adduced by the authors, I am still not convinced that the idea of „conformation capturing” represents a novel concept (see below for detailed discussion) and constitutes the main value of the paper. However, because I don't neglect experimental facts, but only their interpretation, and I really appreciate the intriguing structures of the cages and the huge amount of experimental work on their properties, I can recommend the paper for publication. However, I think that the title and the abstract should be modified according to the current content.

1. The authors mention the energy difference for 1 as one of the arguments in the discussion (: „However, as we calculated the conformation energy of the two conformers of molecule 1 (Supplementary Fig. 9 and Table 1), we found that their energy difference is negligible (ca. 0.1 kJ mol⁻¹).” For a dynamic system, the relative stability of the products matters not of the substrates. Not to mention that for identification of conformers it is the barrier of interconversion, not the relative stability that matters.
2. Size match – the authors argue that the results are not intuitive because the average size of the cleft does not agree with the observed selectivity. However, this is not the average size, but the minimum size is the factor that matters and, in this context, the results are intuitive. The authors themselves admit this „ „ as the maximally stretched amines of CC-1 154 still cannot reach the three formyl groups within each cleft”
3. The sentence „conversion rates of endo- and exo-[1,2,5]diphane’ were different, as the former was generated faster than 211 the latter, leading to a product distribution of 75% endo- and 25% exo-[1,2,5]diphane’, respectively.” is not precise, because it is not the kinetics but thermodynamics that determines the final distribution. It should be corrected.

Reviewer #3 (Remarks to the Author):

Although the authors have satisfactorily addressed most of my previous comments, there are some theoretical points that need a better explanation for a total recommendation of the manuscript in Nature Communications. These points are:

1) According to my suggestions, authors have performed calculations including solvent effects and considering two density functionals with Grimme's dispersion corrections (B3LYP-D3 and M062X-D3). Nevertheless, I do not see why the authors have decided to preserve the results at the B3LYP-D3 level since the energy order using total energies or free energies is similar at the M062X level. A justification should be provided. In Table 1, the units are missing (I guess kJ/mol).

2) The computed torsion potential for the rotation of the carbon-carbon bond of the central benzene ring and the subsequent sp³ carbon atom does not seem to suggest an easy rotation (according to the energetics) and the simple cartoon used in Figure 1, middle-left cannot be appropriate. A clarification of this should be provided. The type of torsion potential, either rigid or fully relaxed should be indicated in the text or supplementary information. Finally and for consistency with other energies, the units in the torsion graph should be in kJ/mol.

3) The reply to my previous comment (comment 6th in the previous letter) related to MD calculations is elusive unsatisfactory. First, the sentences added explain the type of DFT calculations that have been performed over the two Conformers-1 and -2. A call for the MD calculations is appropriate when Figure 1 is explained because, based on those calculations, the cartoon in Figure 1 (middle-left) is created. Second, I keep thinking that the histograms in Supplementary Figure 10 have been able to be interchanged. If we compare the cartoon for Conformer-2 with its histogram there are possible events with distances lower than 0.4 nm, which does not match with the cartoon (distances between 0.6 – 1.3 nm). This should be clarified if there are no mistakes.

4) Again, my previous comment (comment 7th in the previous letter) related to the relative energies between both cages (exo and endo) has not properly addressed. The authors should provide the energy difference between both structures, which can help to see which structure is more stable.

Typos. Page 4 in the revised version. "Optimizations" instead of "Optimizatoins". In the same line, it would be better "...DFT calculations..."

Reviewer 1:

I am pleased to have read the revised version of the manuscript by Professor Zhang and co-workers which has gone through a substantial edit and as a result has been significantly improved.

Answer: First of all, we would like to thank the Reviewer again for carefully reviewing our manuscript and appreciating our research work.

Q1: The PXRD part of the new version of the manuscript has been thoroughly revised and I am glad to see that a correlation between the simulated and recorded powder data now exists. The peak widths of the experimental patterns are rather large considering that single crystals were used in data collection. This could be explained, however, by partial desolvation of the crystals if the capillaries used in data collection were loaded with dried crystals and not together with mother liquor. This small detail, i.e. were the capillaries loaded with or without crystallization solvent, is an important information and should be added in the experimental detail in the supporting information to avoid confusion.

A1: Thanks for the reviewer's comments. The Reviewer is indeed correct. We therefore have revised the experimental details as "Slow evaporation of the settled solution yielded high-quality single crystals, which were then loaded into a quartz glass capillary with a diameter of 0.7 mm without mother liquor", which was highlighted in yellow in Powder X-ray Diffraction section (section 2.6, Page S5) of the Revised Supplementary Information.

Q2: The given densities in single crystal X-ray diffraction data tables (Supplementary Tables 2-6) should be in g/cm³ (not in M g/cm³).

A1: Thanks for the reviewer's careful remark. The density units of the single crystal X-ray diffraction (Supplementary Tables 2-6, Page S24-S28) should be either Mg/m³ or g/cm³, and we therefore used "g/cm³" in the revised version, which are highlighted in yellow in the Revised Supplementary Information.

Q3: Check the graph legend in Supplementary Figure 14, which has the descriptions of the different graphs in wrong order.

A1: Thanks for the reviewer's comments. The graph legend in Supplementary Figure 14 in Page S30 of the Revised Supplementary Information has been amended.

Reviewer 2:

Despite numerous arguments adduced by the authors, I am still not convinced that the idea of conformation capturing” represents a novel concept (see below for detailed discussion) and constitutes the main value of the paper. However, because I don’t neglect experimental facts, but only their interpretation, and I really appreciate the intriguing structures of the cages and the huge amount of experimental work on their properties, I can recommend the paper for publication.

First of all, we would like to thank the Reviewer again for carefully reviewing our manuscript and appreciating our research work.

Q1: However, I think that the title and the abstract should be modified according to the current content. The authors mention the energy difference for **1** as one of the arguments in the discussion (However, as we calculated the conformation energy of the two conformers of molecule **1** (Supplementary Fig. 9 and Table 1), we found that their energy difference is negligible (ca. 0.1 kJ mol⁻¹).” For a dynamic system, the relative stability of the products matters not of the substrates. Not to mention that for identification of conformers it is the barrier of interconversion, not the relative stability that matters.

A1: Thanks for the reviewer’s comments. The reviewer’s opinion is right and we have made the following corrections in the Revised Manuscript.

- 1). The title has been amended as “Diphanes: A Class of Twin-Cavity Cages”.
- 2). The abstract has been revised to emphasize the structure of the twin-cavity cages (diphane) and their properties as follows: Covalent organic cages recently have attracted wide interest in the fields of recognition/separation, sensor, catalysis, etc. A variety of organic cages with different fascinating geometries have been developed during the last decade, but most of them exhibit a single cavity. We envisioned that a cage with a pair of cavities could open the way for the search of novel porous materials with emergent properties and functions. Here, as a proof of concept, we rationally designed a three-dimensional hexaformyl precursor **1**, which exhibits two types of conformers, *i.e.* **Conformer-1** and **-2** with different cleft positions and sizes. Aided by molecular dynamics simulations, we selected two triamino conformation capturers (denoted CC). Small-sized **CC-1** selectively captured **Conformer-1** by matching its cleft size, while large-sized **CC-2** was able to match and capture both conformers. This strategy allowed the formation of two sandglass-shaped and one dumbbell-like compounds with twin cavities, which we coined *diphane*. The self-assembly of the three *diphanes* in turn led to the discovery of supramolecular materials with tunable proton conductivity, which reached up to 1.37×10^{-5} S cm⁻¹,

approximately 10^3 times higher than bulk water. Depending on the configuration of *diphanes*, their conductivity can be tuned by an order of magnitude of 10^4 .

Q2: Size match – the authors argue that the results are not intuitive because the average size of the cleft does not agree with the observed selectivity. However, this is not the average size, but the minimum size is the factor that matters and, in this context, the results are intuitive. The authors themselves admit this “as the maximally stretched amines of CC-1 still cannot reach the three formyl groups within each cleft”

A2: Thanks for the reviewer’s comments. We agree with the reviewer’s opinion that the minimum size of the clefts play a more important role in the formation of *diphanes* with different geometry. It proves we still cannot convince the reviewer with the size-matching strategy as the novelty. We had already changed the focal point and interpretation of the experimental results in the first revision, and therefore have revised the title and abstract in the second revision to comply with the Reviewer’s comment.

Q3: The sentence “conversion rates of endo- and exo-[1,2,5]diphane’ were different, as the former was generated faster than the latter, leading to a product distribution of 75% endo- and 25% exo-[1,2,5]diphane’, respectively.” is not precise, because it is not the kinetics but thermodynamics that determines the final distribution. It should be corrected.

A3: Thanks for the reviewer’s comments. The reviewer’s opinion is correct and the term “faster” is not rigorous enough. We therefore revised the phrase as “The evolution profile illustrates that the consumption of molecule **1** occurred simultaneously with the formation of two *diphane* isomers until the equilibrium, which led to a product distribution of 75% *endo*- and 25% *exo*-[1,2,5]diphane’, respectively.” in Page 10 of the Revised Manuscript.

Reviewer 3:

Although the authors have satisfactorily addressed most of my previous comments, there are some theoretical points that need a better explanation for a total recommendation of the manuscript in Nature Communications.

Answer: First of all, we would like to thank the Reviewer again for carefully reviewing our manuscript and appreciating our research work.

Q1: According to my suggestions, authors have performed calculations including solvent effects and considering two density functionals with Grimme's dispersion corrections (B3LYP-D3 and M062X-D3). Nevertheless, I do not see why the authors have decided to preserve the results at the B3LYP-D3 level since the energy order using total energies or free energies is similar at the M062X level. A justification should be provided. In Table 1, the units are missing (I guess kJ/mol).

A1: Thanks for the reviewer's comments.

1). As per request of the Reviewer in the first revision, we employed M062X-D3, and compared with the results of B3LYP-D3 in the original manuscript. We found that the energy difference of the **Conformer-1** and **-2** calculated with B3LYP-D3 (-0.12 kJ/mol) and M062X-D3 (0.72 kJ/mol) functionals is very similar within the range of experimental error, which confirms their fast interchange in CHCl₃. According to the Grimme's work (*Phys. Chem. Chem. Phys.*, 2011, **13**, 6670–6688), B3LYP-D3 is indeed not the overall applicable functional (which might be the concern of the Reviewer). The authors showed that among all tested 23 hybrids, M062X-D3 is statistically the best of all hybrids. On the other hand, they also observed SCF-convergence problems for all Minnesota functionals, even for simple atomic systems. In our calculations, we also found that some of the optimized relaxed structures through M062X-D3 functional were indeed difficult to converge. Under this consideration, the results reported in this paper are thus all calculated by B3LYP-D3 method in chloroform unless with specific instructions. During the revision, we have provided a justification which is highlighted in yellow in DFT calculations section (starting from Page S18) of the Revised Supplementary Information.

2). We have added the unit of kJ/mol in Supplementary Table 1 of the Revised Supplementary Information.

Q2: The computed torsion potential for the rotation of the carbon-carbon bond of the central benzene ring and the subsequent sp³ carbon atom does not seem to suggest an easy rotation (according to the energetics) and the simple cartoon used in Figure 1, middle-left cannot be appropriate. A clarification of this should be provided. The type of torsion potential, either rigid or fully relaxed should be indicated in the text or

supplementary information. Finally and for consistency with other energies, the units in the torsion graph should be in kJ/mol.

A2: Thanks for the reviewer's comments.

1). Following the reviewer's suggestion, we have differentiated the rotational C-C bonds with letter **a** and **b** in Fig. 1 of the Revised Manuscript, and revised the second sentence of the caption as "The three-dimensional molecule **1** experiences spontaneous interconversion between **Conformer-1** with a pair of upper and lower clefts and **Conformer-2** with a pair of left and right clefts (left column), of which C-C bonds **a** are much more liable to rotate than C-C bonds **b**, as the latter has a higher rotational energy barrier determined by relaxed potential energy scan." Meanwhile, we have also revised the sentences of the Revised Manuscript (line 2, in Page 4) as "molecule **1** exists infinite forms of conformers *via* all possible C-C single bond rotations, of which C-C bonds **a** are much more liable to rotate than C-C bonds **b**, as the latter has a higher rotational energy barrier (Supplementary Fig. 9). It therefore leads to the formation of two types of conformers, **Conformer-1** and **-2** (Fig. 1 and Fig. 2a)."

2). The conformation energies were determined by relaxed rather than rigid potential energy scan. We have revised the caption of Supplementary Figure 9 as "**Supplementary Figure 9.** Variation of conformation energy with 1) the rotation around the inter-ring carbon-carbon bond **a** of the two aryl rings within a peripheral arm and 2) the rotation of the carbon-carbon bond **b** of the central benzene ring and the subsequent sp³ carbon atom. The conformation energies were determined by relaxed potential energy scan". A brief description of the Supplementary Figure 9 has been added in Page S19 of the Revised Supplementary Information.

3). In Supplementary Figure 9 of the Revised Supplementary Information, the units in the torsion graph (left) have been converted to the unit of kJ/mol.

Q3: The reply to my previous comment (comment 6th in the previous letter) related to MD calculations is elusive unsatisfactory. First, the sentences added explain the type of DFT calculations that have been performed over the two Conformers-1 and -2. A call for the MD calculations is appropriate when Figure 1 is explained because, based on those calculations, the cartoon in Figure 1 (middle-left) is created. Second, I keep thinking that the histograms in Supplementary Figure 10 have been able to be interchanged. If we compare the cartoon for Conformer-2 with its histogram there are possible events with distances lower than 0.4 nm, which does not match with the cartoon (distances between 0.6 – 1.3 nm). This should be clarified if there are no mistakes.

A3: Thanks for the reviewer's comments.

1. Following the reviewer's suggestion, we revised the sentence of the Revised Manuscript (line 2 from the bottom, in Page 3) as "As briefly mentioned above, the region with the probability of containing formyl groups varies within the cleft of each conformer were determined by molecular dynamics simulations (see detailed MD simulation in Supplementary Fig. 10), which showed that..."

2. First of all, it's our negligence that we put the two histograms in a wrong order, and the Supplementary Figure 10 therefore has been revised in Page S21 of the Revised Supplementary Information. Second, the histograms could be indeed interchanged. For easy understanding, we artificially classified two types of formyl group activity regions, which correspond to **Conformer-1** and **-2**, respectively. During the simulations of the formyl groups' distributions, we first counted all the formyl groups (red dots) located inside the regions of **Conformer-1** and **-2**, respectively, and then mapped the corresponding histograms.

Q4: Again, my previous comment (comment 7th in the previous letter) related to the relative energies between both cages (exo and endo) has not properly addressed. The authors should provide the energy difference between both structures, which can help to see which structure is more stable.

A4: Thanks for the reviewer's comments. According the reviewer's suggestion, we have provided the relative configuration energy difference between *diphanes* in Supplementary Figure 11 in Page S22 of the Revised Supplementary Information. The results showed that *endo*-[1,2,4]diphane is 51.02 kJ/mol more stable than *exo*-[1,2,4]diphane, and *endo*-[1,2,5]diphane is 23.86 kJ/mol more stable than *exo*-[1,2,5]diphane, which correlate well with the experimental yields obtained for the resulting *diphanes*. This has also been explained and highlighted in yellow in the following paragraphs in Page S22-23 of the Revised Supplementary Information.

Q5: Typos. Page 4 in the revised version. "Optimizations" instead of "Optimizatoins". In the same line, it would be better "...DFT calculations..."

A5: Thanks for the reviewer's careful remark. The corresponding description (line 2 from the bottom, Page 3) has been revised as "As briefly mentioned above, the region with the probability of containing formyl groups varies within the cleft of each conformer were determined by molecular dynamics simulations (see detailed MD simulation in Supplementary Fig. 10) showed that..." in the Revised Manuscript. As the whole sentence has been modified, we don't use "optimization" and "DFT calculations" anymore in the newly revised version.

Changes List for Manuscript NCOMMS-21-02237 and Supplementary Information

➤ In manuscript

1. The title has been revised as “Diphanes: A Class of Twin-Cavity Cages” in the Revised Manuscript.
2. The abstract has been amended in the Revised Manuscript, which are highlighted in yellow.
3. In Fig. 1 of the Revised Manuscript, label **a** and **b** have been added on the figure of molecule **1** to differentiate C-C bonds with different rotational flexibility and the caption has been revised accordingly.
4. The first paragraph of the Results part has been revised in the Revised Manuscript, which are highlighted in yellow.
5. The 5th line of the caption in Fig. 2, the word “tempeatures” has been revised as “temperature” in the Revised Manuscript.
6. The 4th line of the caption in Fig. 5, the word “crystallograpy” has been revised as “crystallography” in the Revised Manuscript.
7. In Page 8, the 9th line from the bottom, the word “possiblity” has been revised as “possibility” in the Revised Manuscript.
8. In Page 9, the 8th line, the word “disapperance” has been amended as “disappearance” in the Revised Manuscript.
9. The 2th line of the caption in Fig. 6, the word “simutaniouly” has been revised as “simultaneously” in the Revised Manuscript.
10. In Page 10, the 5th line, the sentence has been amended as “The evolution profile illustrates that the consumption of molecule **1** occurred simultaneously with the formation of two *diphane* isomers until the equilibrium, which has a product distribution of 75% *endo*- and 25% *exo*-[**1,2,5**]diphane’, respectively.” in the Revised Manuscript.
11. In Page 12, the last word of the first paragraph “conducition” has been revised as “conduction” in the Revised Manuscript.
12. In Page 13, the first line, the word “retrorespective” has been revised as “retrospective” in the Revised Manuscript.
13. In Page 13, line 3 from the bottom of the second paragraph, the word “effeciency” has been revised as “efficiency” in the Revised Manuscript.
14. In Page 13, line 3 from the bottom of the second paragraph, the word “effeciency” has been revised

as “efficiency” in the Revised Manuscript.

15. In Page 13, the last line of the second paragraph, the word “the” has been deleted in the Revised Manuscript.

➤ **In Supplementary Information**

1. The title has been revised as “Diphanes: A Class of Twin-Cavity Cages” in the Revised Supplementary Information.
2. In Page S5, the 6th line, the sentence has been revised as “Slow evaporation of the settled solution yielded high-quality single crystals, which were then loaded into a 0.7 mm diameter quartz glass capillary without mother liquor.” in the Revised Supplementary Information.
3. In Page S18, a new paragraph has been added and highlighted in yellow in the Revised Supplementary Information.
4. In Page S18, a new reference S14 has been inserted and some of the related references’ order has been adjusted accordingly, which are highlighted in yellow in the Revised Supplementary Information.
5. In Page S19, the Supplementary Figure 9 and the caption have been amended, meanwhile a brief description of the Supplementary Figure 9 has been added in the Revised Supplementary Information.
6. In Page S21, the Supplementary Figure 10 has been revised in the Revised Supplementary Information.
7. In Page S22, the Supplementary Figure 11 has been revised and its description has also been amended, which are highlighted in yellow in the Revised Supplementary Information.
8. The density units of the single crystal X-ray diffraction (supplementary tables 2-6) have been revised as “g/cm³”, which are highlighted in yellow in the Revised Supplementary Information.
9. In Page S30, the graph legend in Supplementary Figure 14 has been amended in the Revised Supplementary Information.

REVIEWER COMMENTS

Reviewer #3 (Remarks to the Author):

Now, all my previous comments have been satisfactorily addressed and I would recommend the publication of this manuscript in Nature Communications.